



# In-orbit Earth reflectance validation of TROPOMI on board the Sentinel-5 Precursor satellite

Lieuwe G. Tilstra[1], Martin de Graaf[1], Ping Wang[1], and Piet Stammes[1]

[1]Royal Netherlands Meteorological Institute (KNMI), De Bilt, The Netherlands

**Correspondence:** L. G. Tilstra (tilstra@knmi.nl)

**Abstract.** The goal of the study described in this paper is to determine the accuracy of the radiometric calibration of the TROPOMI instrument in-flight, using its Earth radiance and solar irradiance measurements, from which the Earth reflectance is determined. The Earth reflectances are compared to radiative transfer calculations. We restrict ourselves to clear-sky observations as these are less difficult to model than observations containing clouds and/or aerosols. The limiting factor in the radiative

transfer calculations is then the knowledge of the surface reflectance. We use OMI and SCIAMACHY surface Lambertian-equivalent reflectivity (LER) information to model the reflectivity of the Earth's surface. This Lambertian, non-directional description of the surface reflection contribution results in a relatively large source of uncertainty in the calculations. These errors can be reduced significantly by filtering out geometries for which we know that surface LER is a poor approximation of the real surface reflectivity. This filtering is done by comparing the OMI/SCIAMACHY surface LER information to MODIS

surface BRDF information.

      We report calibration accuracies and errors for 21 selected wavelength bands between 328 and 2314 nm, located in TROPOMI spectral bands 3–7. All wavelength bands show good linear response to the intensity of the radiation and negligible offset problems. Reflectances in spectral bands 5 and 6 (wavelength bands 670 to 772 nm) have a good absolute agreement with the simulations, showing calibration errors on the order of 0.01 or 0–3%. Trends over the mission lifetime, due to instrument

degradation, are studied and found to be negligible at these wavelengths. Reflectances in bands 3 and 4 (wavelength bands 328 to 494 nm), on the other hand, are found to be affected by serious calibration errors, on the order of 0.004–0.02 and ranging between 6% and 10%, depending on the wavelength. The TROPOMI requirements (of 2% maximal deviation) are not met in this case. Trends due to instrument degradation are also found, being strongest for the 328-nm wavelength band, and almost absent for the 494-nm wavelength band.

The validation results obtained for TROPOMI spectral band 7 show behaviour that we cannot fully explain. As a result, these results call for more research and different methods to study the calibration of the reflectance. It seems plausible, though, that the reflectance for this particular band is underestimated by about 6%. A table is provided containing the final results for all 21 selected wavelength bands.



# 1 Introduction

Satellite monitoring of atmospheric composition has evolved a lot over the last decades. Continuous monitoring started out in the year 1978 with the launch of the first Total Ozone Mapping Spectrometer (TOMS) instrument. The TOMS instrument was designed to measure the Earth reflectance at six wavelengths between 310 and 380 nm (Heath et al., 1975). With this

limited spectral information, it was possible to retrieve ozone, $SO_2$, and various other properties globally on a daily basis. The spectrometer Global Ozone Monitoring Experiment (GOME) (Burrows et al., 1999), launched in 1995, was the first satellite instrument recording high-resolution spectra. The spectra were measured with a spectral resolution of 0.2–0.3 nm over the spectral range between 240 and 790 nm. Retrieval of atmospheric species from the spectra was achieved using the differential optical absorption spectroscopy (DOAS) method (Platt, 1994).

Since then, many more spectrometers have been launched, including Scanning Imaging Absorption Spectrometer for Atmospheric Chartography (SCIAMACHY) (Bovensmann et al., 1999), launched in 2002, Ozone Monitoring Instrument (OMI) (Levelt et al., 2006), launched in 2004, the GOME-2A/B/C instruments on the MetOp series of satellites, launched in 2006, 2012, and 2018, respectively, and Tropospheric Monitoring Instrument (TROPOMI), launched in October 2017 and the focus of this paper. Each of these satellite instruments represents an improvement with respect to its predecessors in terms of spatial

resolution, spectral coverage, spectral resolution, and/or performance. Using the spectral coverage and resolution offered by the spectrometers, many atmospheric species can be retrieved successfully, including ozone, $SO_2$, $NO_2$, BrO, $CH_2O$, $H_2O$, CO, $CH_4$, as well as surface, cloud, and aerosol properties.

The quality of the retrieved properties is depending in one way or another on the quality of the radiometric calibration of the satellite instrument. For DOAS retrievals the absolute calibration of the reflectances is not so important, because any constant

radiometric error cancels out in the DOAS method (Platt and Stutz, 2008). However, for many products the retrieval is not based on the DOAS method. Examples are the retrieval of ozone profiles (Hasekamp et al., 2002; Liu et al., 2010; van Peet et al., 2014; Shah et al., 2018), absorbing aerosol index (Herman et al., 1997; Torres et al., 1998; de Graaf et al., 2005), surface reflectivity (Herman and Celarier, 1997; Koelemeijer et al., 2003; Kleipool et al., 2008; Tilstra et al., 2017), and cloud and aerosol properties (Wang et al., 2008, 2012; Joiner et al., 2012; Torres et al., 2013; Lelli et al., 2014). For all these products the

retrieval codes do rely on a correct absolute calibration of the reflectances.

The radiometric calibration of a satellite instrument starts on-ground, in a laboratory, with extensive characterisation of the instrument's response to the exposure of radiation. However, the conditions in space are different from the conditions in the laboratory environment. Also, before, during, and after launch the instrument can undergo instrumental, electronic or optical degradation. Once the satellite is in orbit and the instrument is switched on in-flight monitoring starts. This in-flight monitoring

is primarily based on measuring the solar irradiance signal on a daily basis, to use it as a reference under the assumption that the Sun is a stable source. These daily measurements of the solar signal can point to shortcomings in the pre-flight radiometric calibration and they can reveal changes in the radiometric response of the instrument due to instrument degradation. They can be used to derive corrections on the pre-flight calibration of the solar irradiance.



However, this only corrects the solar irradiance, not the Earth radiance. The assumption that both signals degrade in the same manner is not justified (see e.g. van der A et al., 2002; Tilstra et al., 2012). As a result of this, the reflectance, which is a ratio of the Earth radiance and the solar irradiance, may not be corrected properly. To address this issue, in-flight calibration of the reflectance is needed, as has been done in the past in various ways, such as by intercomparison with other satellite

instruments (Acarreta and Stammes, 2005; Kokhanovsky et al., 2007; Tilstra and Stammes, 2006, 2007; Jourdan et al., 2007), or by comparison with radiative transfer calculations (van Soest et al., 2005; Jaross and Warner, 2008; Tilstra et al., 2005, 2014; Cai et al., 2012).

In this paper we study and validate the radiometric calibration of the TROPOMI instrument by comparing the reflectance measured in-flight with radiative transfer calculations. This is done for clear-sky land and ocean scenes for wavelengths ranging

from 328 to 2314 nm. The difference compared to earlier studies is that these were restricted to the UV wavelength range (below 400 nm). The reason for this was that for longer wavelengths the influence of the surface albedo on the reflectance is higher while at the same time the uncertainty in the input surface albedo for the simulations increases. In this study we perform the analyses for TROPOMI bands 3–7, i.e. for the much larger UV-VIS-NIR-SWIR wavelength range, using the best possible relevant surface albedo input available at this time.

The outline of this paper is as follows. Section 2 provides a brief description of the TROPOMI instrument. Section 3 describes the radiative transfer modelling and the most important input parameters, such as the surface reflectivity. Section 4 describes the approach that was followed, including the selection of the wavelength bands, the cloud and aerosol screening, and the selection of scenes to analyse. Section 5 discusses ways to reduce the impact of surface albedo errors. Section 6 presents the results, and draws conclusions about the radiometric calibration of TROPOMI. Trend analysis, to study instrument degradation, is also part

of this section. The paper ends with the most important conclusions in Sect. 7.

## 2 Description of TROPOMI

### 2.1 Instrument description

The TROPOMI instrument (Veefkind et al., 2012) is the only instrument on board the Sentinel-5 Precursor (S5P) satellite. The S5P satellite was launched on 13 October 2017 and was put into a near-polar, Sun-synchronous orbit with a mean altitude

of 824 km above the Earth's surface, an equator crossing time of 13:30 LT, and a repeat cycle of 17 days. TROPOMI is the successor of OMI (Levelt et al., 2006), which is kept in a very similar orbit to allow accurate intercomparisons and cross-validation. OMI has proven itself to be important for many advances in the field of satellite remote sensing (Levelt et al., 2018). TROPOMI is meant to expand on these successes, and to fill the gap that exists between the SCIAMACHY mission, which ended in April 2012 with the loss of the Envisat platform, and the future Sentinel-5 mission, which is planned for launch

in 2022.

TROPOMI, like its predecessor OMI, is a nadir-looking spectrometer that observes the spectral domain in the across-track dimension in one go using a two-dimensional detector. However, while OMI only observes the ultraviolet-visible wavelength range (270–500 nm), TROPOMI is able to observe the ultraviolet-visible (UV-VIS, 267–499 nm), near-infrared (NIR, 661–





786 nm), and shortwave infrared (SWIR, 2300–2389 nm) wavelength ranges. TROPOMI is equipped with four detectors, which each host two spectral bands. The characteristics of these eight spectral bands are found in Table 1. The extension of the wavelength range (compared to OMI) allows the retrieval of additional trace gases such as $CO$ and $CH_4$ and better retrieval of cloud and aerosol information using the $O_2$-A and $O_2$-B absorption bands.

Another improvement with respect to OMI is the high spatial resolution of TROPOMI. The TROPOMI footprint size was $7.2 \times 3.6$ km$^2$, but since 6 August 2019 this has been reduced to $5.6 \times 3.6$ km$^2$. TROPOMI observes the sunlit side of the Earth when S5P is in the ascending part of its orbit. The orbit swath is 2600 km wide, allowing global coverage in one day. The TROPOMI commissioning phase ended on 30 April 2018. This is officially the date from which the data can be used. The current level-1 version is 1.0. Further information about the instrument and the derived products can found in Veefkind et al.

(2012) and Kleipool et al. (2018).

## 2.2   On-board calibration

For the sake of on-board calibration TROPOMI is equipped with a number of internal light sources. LED strings are mounted next to each of the four detectors (DLED), to monitor the performance of the detectors themselves. For the visible wavelength range a LED (called common LED, CLED) is placed in the calibration unit, to monitor the throughput of the internal light path

from the calibration unit to the detectors (via telescope, slit, and spectrometers). Additionally, a white light source (WLS) is installed in this calibration unit. All these light sources are meant to provide stable signals for the purpose of monitoring the instrument's throughput (Ludewig et al., 2020). Note that the CLED and WLS measurements can only provide information for part of the internal light paths that are used for the radiance and solar irradiance measurements.

**Table 1.** Characteristics of the TROPOMI detectors and spectral bands. The eight spectral bands are located on the four detectors. The design spectral coverage of the detectors and the full spectral coverage of the spectral bands are both listed. The spatial sampling in the along track direction was changed from the indicated 7.2 km to 5.6 km on 6 August 2019, at the start of orbit 9388.

| Detector | UV | | UV-VIS | | NIR | | SWIR | |
|---|---|---|---|---|---|---|---|---|
| Spectral coverage [nm] | 270–320 | | 320–495 | | 675–775 | | 2305–2385 | |
| Spectral band | 1 | 2 | 3 | 4 | 5 | 6 | 7 | 8 |
| Full spectral coverage [nm] | 267–300 | 300–332 | 305–400 | 400–499 | 661–725 | 725–786 | 2300–2343 | 2343–2389 |
| Spectral resolution [nm] | 0.47 | 0.46 | 0.51 | 0.50 | 0.34 | 0.34 | 0.23 | 0.23 |
| Spectral sampling ratio | 7.2 | 7.1 | 2.6 | 2.5 | 2.7 | 2.8 | 2.4 | 2.4 |
| Pixel size across track [km] | 29.6 | 3.6 | 3.6 | 3.6 | 3.6 | 3.6 | 7.1 | 7.1 |
| Pixel size along track [km] | 7.2 | 7.2 | 7.2 | 7.2 | 7.2 | 7.2 | 7.2 | 7.2 |
| (after 6 August 2019:) | 5.6 | 5.6 | 5.6 | 5.6 | 5.6 | 5.6 | 5.6 | 5.6 |




Daily solar measurements provide another reference, and from these the stability of the solar irradiance over time can be assessed. The absolute radiometric calibration of the solar irradiance can be verified easily because the Sun is a relatively stable light source. For the radiance measurements, other techniques have to be applied.

## 3 Radiative transfer modelling

### 3.1 Theoretical background

In this paper, the Earth reflectance is defined as

$$R = \frac{\pi I}{\mu_0 E} \ . \tag{1}$$

The quantity $I$ is the Earth radiance at the top-of-atmosphere (TOA), in units $\mathrm{Wm^{-2}sr^{-1}nm^{-1}}$. The quantity $E$ is the solar irradiance at the TOA, perpendicular to the incoming solar beam, in units $\mathrm{Wm^{-2}nm^{-1}}$. The parameter $\mu_0$ is defined as $\mu_0 = \cos\theta_0$, where $\theta_0$ is the solar zenith angle. For the viewing direction we use a similar definition, namely $\mu = \cos\theta$, with $\theta$ the viewing zenith angle. The viewing and solar azimuth angles are symbolised by $\phi$ and $\phi_0$, respectively.

For the radiative transfer calculations we restrict ourselves to clear-sky scenes and impose Lambertian surface reflection. According to Chandrasekhar (1960) the following relationship is then valid:

$$R(\mu, \mu_0, \phi, \phi_0, A_\mathrm{s}) = R^0(\mu, \mu_0, \phi - \phi_0) + \frac{A_\mathrm{s} T(\mu, \mu_0)}{1 - A_\mathrm{s} s^\star} \tag{2}$$

In Eq. (2), the quantity $R^0$ is the so-called path reflectance, which represents the purely atmospheric contribution to the reflectance. In other words, it corresponds to the reflectance of the Rayleigh atmosphere when it is bounded below by a black surface that does not reflect any of the incoming radiation. The second term represents the surface contribution to the reflectance. This terms is determined by the surface albedo $A_\mathrm{s}$, by the total transmission of the atmosphere $T$, and by $s^\star$, the spherical albedo of the atmosphere for light coming from below. The quantities $R^0$, $T$ and $s^\star$ can be determined from radiative transfer calculations.

### 3.2 Radiative transfer calculations

For the radiative transfer calculations we make use of the "Doubling-Adding KNMI" (DAK) radiative transfer code (de Haan, 1987; Stammes, 2001). This code can calculate the monochromatic Earth reflectance taking polarisation into account. It can handle molecular scattering, scattering and absorption by clouds and aerosols, and absorption by various trace gases. DAK employs Lambertian surface reflection and simulates a pseudo-spherical atmosphere. We used version 3.2.0. The calculations were performed for the 21 wavelength bands listed in Table 2. We used a standard Mid-Latitude Summer (MLS) atmospheric profile (Anderson et al., 1986) and included absorption by ozone, $NO_2$, and $O_2$-$O_2$. For some of the wavelength bands we additionally included absorption by oxygen (697, 712, 758, and 772 nm) and/or water vapour (697 and 712 nm). For these wavelength bands monochromatic calculations do not suffice and spectral calculations are in order. Table 2 indicates when this is the case. Clouds and aerosols were not included in the simulations, because in this study we only focus on clear-sky scenes.





**Table 2.** Definition of the wavelength bands and of the way the radiative transfer calculations were performed.

| Wavelength band | 328 | 335 | 340 | 354 | 367 | 380 | 388 | 402 | 416 | 425 | 440 |
|---|---|---|---|---|---|---|---|---|---|---|---|
| Instrument channel | 3 | 3 | 3 | 3 | 3 | 3 | 3 | 4 | 4 | 4 | 4 |
| Central wavelength (nm) | 328.0 | 335.0 | 340.0 | 354.0 | 367.0 | 380.0 | 388.0 | 402.0 | 416.0 | 425.0 | 440.0 |
| Bandwidth (nm) | 1.0 | 1.0 | 1.0 | 1.0 | 1.0 | 1.0 | 1.0 | 1.0 | 1.0 | 1.0 | 1.0 |
| Spectral/monochromatic | S | M | M | M | M | M | M | M | M | M | M |
| Ozone absorption | + | + | + | + | + | + | + | + | + | + | + |
| $NO_2$ absorption | + | + | + | + | + | + | + | + | + | + | + |
| $O_2$-$O_2$ absorption | + | + | + | + | + | + | + | + | + | + | + |
| Oxygen absorption | – | – | – | – | – | – | – | – | – | – | – |
| Water vapour absorption | – | – | – | – | – | – | – | – | – | – | – |
| Wavelength band | 463 | 494 | 670 | 685 | 697 | 712 | 747 | 758 | 772 | 2314 | |
| Instrument channel | 4 | 4 | 5 | 5 | 5 | 5 | 6 | 6 | 6 | 7 | |
| Central wavelength (nm) | 463.0 | 494.0 | 670.0 | 685.0 | 696.97 | 712.7 | 747.0 | 758.0 | 772.0 | 2314.0 | |
| Bandwidth (nm) | 1.0 | 1.0 | 1.0 | 1.0 | 0.3 | 0.3 | 1.0 | 1.0 | 1.0 | 0.5 | |
| Spectral/monochromatic | M | M | M | M | S | S | M | S | S | M | |
| Ozone absorption | + | + | + | + | + | + | + | + | + | + | |
| $NO_2$ absorption | + | + | + | + | + | + | + | + | + | + | |
| $O_2$-$O_2$ absorption | + | + | + | + | + | + | + | + | + | + | |
| Oxygen absorption | – | – | – | – | + | + | – | + | + | – | |
| Water vapour absorption | – | – | – | – | + | + | – | – | – | – | |

The reflectance calculations are performed using spectral band integration or monochromatically. For all wavelength bands absorption by ozone, $NO_2$, and $O_2$-$O_2$ is included.
Absorption by oxygen and/or water vapour is included for only some of the wavelength bands.

### 3.3 Look-up tables

For each of the 21 wavelength bands look-up tables (LUTs) were created. These were calculated for 7 ozone column values
(50, 200, 300, 350, 400, 500, and 650 DU), for 10 surface heights (0–9 km), for two water vapour columns (0 and $4\,\mathrm{g/cm^2}$),
and for 42 non-equidistant values for each of the zenith angles $\mu$ and $\mu_0$. For the path reflectance $R^0$ we make use of a
5   Fourier expansion in terms of the relative azimuth angle $\phi - \phi_0$, which ends after only three terms because the simulations are
performed for clear-sky Rayleigh atmospheres over a completely black surface (Hovenier et al., 2004). That is to say,

$$R^0 = a_0(\mu, \mu_0) + \sum_{i=1}^{2} 2a_i(\mu, \mu_0) \cos i(\phi - \phi_0) \,. \tag{3}$$

The reflectances themselves are not stored in the LUTs but the Fourier coefficient $a_0$, $a_1$, and $a_2$. This has the advantage that
the dependence on the relative azimuth angle $\phi - \phi_0$ can be handled analytically by Eq. (3). The quantities $T$ and $s^\star$ are also



stored in the LUTs. The LUTs therefore only contain the parameters $a_0$, $a_1$, $a_2$, $T$, and $s^\star$. This is sufficient to calculate the reflectance $R$ using Eqs. (3) and (2).

### 3.4 Surface albedo input

Scenes with clouds and/or large amounts of aerosol are excluded in this study so the largest source of errors in the radiative
transfer calculations is the uncertainty in the surface albedo (Tilstra et al., 2005; Cai et al., 2012). We use the OMI and SCIA-MACHY surface LER databases (Kleipool et al., 2008; Tilstra et al., 2017) to provide surface albedo input to the simulations. These databases contain monthly climatological database grids with a spatial resolution of $0.5° \times 0.5°$. The OMI surface LER database, covering the wavelength range 328–499 nm, is used for the wavelengths up to 500 nm. The SCIAMACHY surface LER database, covering the wavelength range 328–2314 nm, is used for the longer wavelengths. The surface albedo contained
in these databases is the Lambertian-equivalent reflectivity (LER) of the surface. Adopting Lambertian surface reflection can cause errors especially for the longer wavelengths. For example, neglect of the directional dependence can lead to errors of a factor of two or more in the surface reflectance of vegetated surfaces (Lorente et al., 2018).

Alternatively, we can make use of MODIS surface BRDF (Gao et al., 2005) as input. Strictly speaking, this is not correct, because the DAK radiative transfer model expects Lambertian surface albedo input. Also, six of the seven MODIS bands for
which BRDF parameters are available are outside the TROPOMI spectral coverage. However, MODIS BRDF can be used to filter out observations for which the Lambertian OMI and SCIAMACHY surface LER databases are not able to provide adequate surface albedo input, see Sect. 5.2. The MODIS product that we use for this is the daily global MODIS MCD43C1 product (Schaaf and Wang, 2015). The spatial resolution of this product is $0.05° \times 0.05°$.

### 3.5 Other input parameters

Other input parameters needed for the calculations are surface height, total ozone column, and water vapour column. The surface height and total ozone column are already available in the TROPOMI UVAI product (Stein Zweers, 2018), and are simply copied. The surface height therefore originates from the GMTED2010 (Global Multi-resolution Terrain Elevation Data) surface elevation database and the total ozone column from ECMWF (European Centre for Medium-Range Weather Forecasts) model data. The water vapour column is taken from daily CAMS (Copernicus Atmosphere Monitoring Service) fields.

## 4 Approach

### 4.1 Wavelength bands

The reflectance validation described in this paper was performed for a selection of 21 wavelength bands (see Table 2 for a list of the bands and their specifications). Figure 1 presents three reflectance spectra measured by TROPOMI over the Sahara desert, the Amazonian rainforest, and the Atlantic ocean. All three spectra were recorded on 20 February 2018 and in each case there
were no clouds present at the time of measurement. The black dotted lines indicate the positions of the 21 wavelength bands





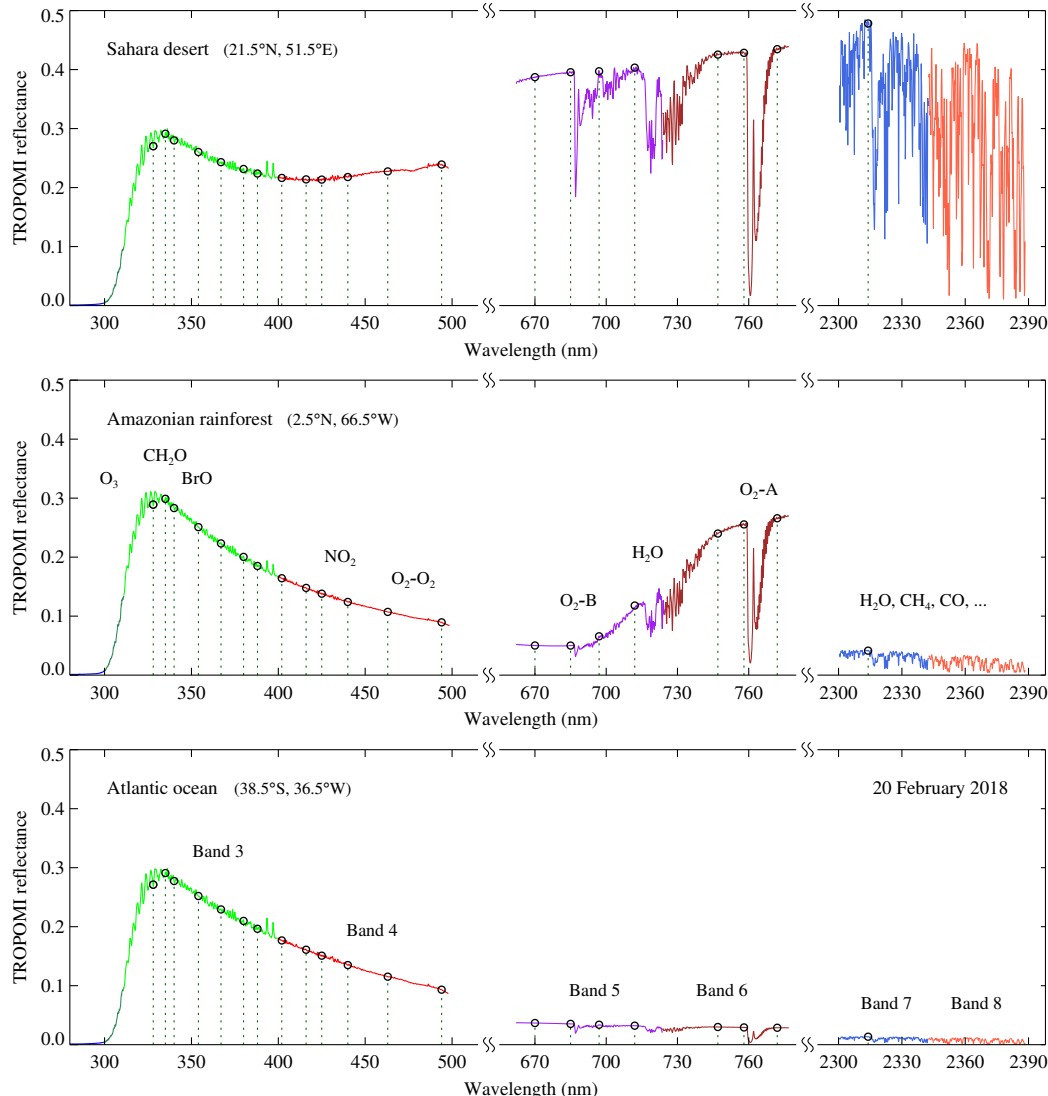

**Figure 1.** Reflectance spectra measured by TROPOMI on 20 February 2018 while observing the Sahara desert, the Amazonian rainforest, and the Atlantic ocean. In all three cases there was almost no cloud presence in the observed scene. Each TROPOMI spectral band was given an own colour. The black dotted lines indicate the position of the 21 wavelength bands that were selected for this study. In the middle panel, absorption bands are indicated by labels of the corresponding trace gases.

that were defined. These wavelength bands cover the TROPOMI spectral range as much as possible while positioned such that there is minimal interference from the surrounding absorption bands.

Nevertheless, for the wavelength bands at 697 and 712 nm some impact of water vapour absorption is present. From Fig. 1 it can be inferred that the impact is relatively small. In spite of this, the dependence on water vapour column is taken into account



in the LUTs, as explained in Sect. 3.2 and 3.3. Below 325 nm the dependence of the reflectance on the (shape of the) ozone profile sets in (Tilstra et al., 2005; Cai et al., 2012). This prevents using a LUT approach and requires knowledge of the ozone profile. This can in principle be handled (e.g., van Soest et al., 2005) but it would complicate matters and it would increase the computational effort (and time needed) by the radiative transfer code considerably. As a result, the shortest wavelength band

5    listed in Table 2 is the one positioned at 328 nm, and none of the wavelength bands are located in spectral bands 1 or 2.

For TROPOMI spectral band 8 no suitable wavelength band can be defined because of strong absorption in the entire wavelength range, predominantly by water vapour. The reflectance validation study is, in conclusion, limited to spectral bands 3–7. Note that the wavelength band at 670 nm is situated outside the design spectral coverage of the NIR detector. This is not a problem because the detectors still perform well a fair bit outside the wavelength range for which they were designed.

## 4.2 Cloud screening

Cloud filtering is performed on the basis of the S5P NPP-VIIRS cloud information product. This product is based on measurements performed by the Visible Infrared Imaging Radiometer Suite (VIIRS) instrument on board the Suomi National Polar-orbiting Partnership (Suomi NPP) satellite. The Suomi NPP satellite is kept in an orbit very similar to that of S5P but with a headstart of about 3 minutes, resulting in a small time difference between the observations made from the two plat-

15   forms. The S5P NPP-VIIRS product provides cloud information, measured by VIIRS, for each of the TROPOMI footprints. It reports, amongst other things, the number of VIIRS observations which were confidently clear ($N_{\mathrm{c.clr}}$), probably clear ($N_{\mathrm{p.clr}}$), probably cloudy ($N_{\mathrm{p.cld}}$), and confidently cloudy ($N_{\mathrm{c.cld}}$) (Siddans, 2016).

We define a geometrical cloud fraction $c_{\mathrm{f}}$ as the number of (confidently plus probably) cloudy S5P-VIIRS observations divided by the total number of S5P-VIIRS observations:

$$c_{\mathrm{f}} = \frac{N_{\mathrm{c.cld}} + N_{\mathrm{p.cld}}}{N_{\mathrm{c.clr}} + N_{\mathrm{p.clr}} + N_{\mathrm{c.cld}} + N_{\mathrm{p.cld}}} \tag{4}$$

The thresholds that are used to filter out cloudy scenes are defined in Sect. 4.4.

## 4.3 Aerosol screening

Aerosol screening is performed using the Absorbing Aerosol Index (AAI) (Torres et al., 1998; de Graaf et al., 2005). We use a relatively high threshold value of 2 index points. This only removes scenes containing the highest concentrations of absorbing

25   aerosol. We do not use the S5P AAI product (Stein Zweers, 2018), but instead calculate the AAI ourselves using the method described in Tilstra et al. (2012). Although the two methods are very comparable, in Sect. 4.4 observations are grouped in boxes and combined, and it is better to calculate the AAI from the average reflectances rather than to average the AAI values. Additionally, we inserted corrections to the reflectances which were fed to the module that calculated the AAI values. This is to prevent offsets and trends that would otherwise be present in the AAI.



## 4.4 Scene selection

The selection of suitable clear-sky scenes is conducted in a way best explained by Fig. 2. The Earth's surface is represented by a $1° \times 1°$ latitude/longitude grid and in Fig. 2 a small part of this grid is shown. The decision to scale down to $1° \times 1°$ latitude/longitude boxes was made with the available surface albedo databases in mind. The course-resolution OMI and

SCIAMACHY surface albedo databases (see Sect. 3.4) are not able to provide realistic surface reflectance input beyond the spatial resolution defined by their grids. The small dots in Fig. 2 represent TROPOMI measurement footprints. We look for $1° \times 1°$ latitude/longitude boxes, such as the one shown in the middle, which are sufficiently cloud-free. This is done by first calculating the geometrical cloud fraction determined from all S5P NPP-VIIRS footprints inside the box, using Eq. (4). The box as a whole is considered cloud-free if this cloud fraction is below 0.03. After that, the geometrical cloud fraction is

determined for the surrounding $3° \times 3°$ latitude/longitude box. If this cloud fraction is below 0.05, then the middle box is not only considered to be cloud-free, but also to be unaffected by cloud shadows from clouds in neighbouring boxes.

The next step is the calculation of the radiance spectrum for the middle box. This is done for each spectral band separately. Co-registration problems that exist between spectral bands of different detectors (Kleipool et al., 2018) are removed automatically because for each spectral band only the footprints that fall inside the box are combined. After that, the Earth reflectance is

determined using Eq. (1). Examples of spectra are shown in Fig. 1. Next, the band reflectances belonging to the 21 wavelength bands are determined. These correspond to the black circles in Fig. 1.

From the band reflectances at 340 and 380 nm the AAI is calculated. If the AAI is lower than 2 then the box is considered aerosol-free, otherwise the box is skipped. Other reasons to skip a box are inhomogeneity (i.e. the box contains both land and water), snow/ice in the box, if the box is too close to the polar regions (absolute latitudes above $60°$), if the solar zenith angle is

above $75°$, if the viewing zenith angle is above $40°$, or if the observations are affected by sun glint or solar eclipse events. The number of boxes found this way amounts to about a thousand per day. This corresponds to about 2–3% of all latitude/longitude boxes between $60°$S and $60°$N. This percentage is in agreement with the percentage that is to be expected for boxes with an area of $\sim$10,000 km$^2$ (Krijger et al., 2007). Figure 3 presents an example of the location and frequency of clear-sky boxes that were collected over the period from May 2018 until February 2020.

The choice of only accepting boxes for which the viewing zenith angle is below $40°$ is also related to the surfaced albedo databases. We expect the OMI database to be mostly representative for the near-nadir viewing geometries, because the algorithm that was used to retrieve the OMI (and SCIAMACHY) surface LER is in most cases focused on the lowest scene reflectances (Kleipool et al., 2008; Tilstra et al., 2017). The SCIAMACHY surface LER database is also mostly representative for the near-nadir viewing geometries, for the simple fact that in the SCIAMACHY orbit swath the viewing zenith angles go

up to $40°$ at most.

After the scene selection and the calculation of the 21 measured band reflectances these band reflectances are simulated in the manner described in Sect. 3. The last step, reduction of the impact of surface albedo errors, is discussed in Sect. 5.



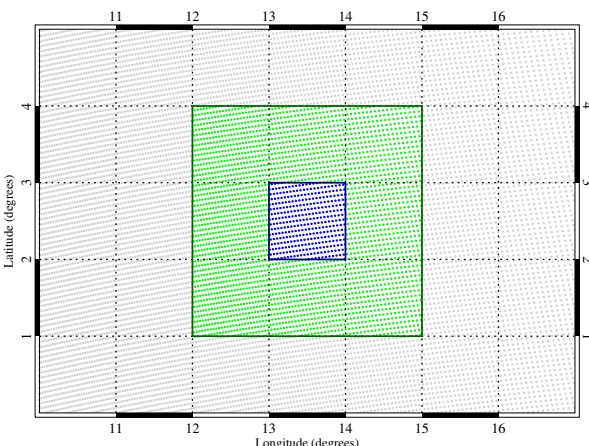

**Figure 2.** Illustration of the approach followed in this paper. The TROPOMI measurement footprints are represented by small dots. We focus, however, on the $1° \times 1°$ latitude/longitude box in the middle. If the geometrical cloud fraction derived from the S5P NPP-VIIRS cloud information of the blue footprints is below 0.03 then the NPP-VIIRS geometrical cloud fraction is determined for the surrounding $3° \times 3°$ latitude/longitude box. If this cloud fraction is below 0.05 then the middle box is considered cloud-free. If it is also considered aerosol free then the TROPOMI reflectance spectrum is calculated for the middle box. This is done for each of the spectral bands. More detailed information is provided in the main text.

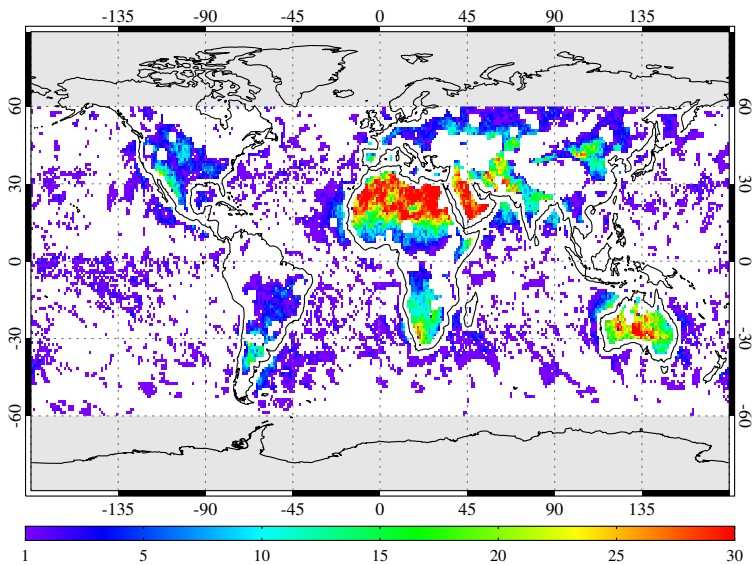

**Figure 3.** Location and frequency of the $1° \times 1°$ cloud-free latitude/longitude boxes that were selected by the algorithm every two weeks for the period from May 2018 until February 2020.



## 5 Methods for reducing surface albedo errors

### 5.1 Surface albedo errors

As explained in Sect. 3.4, the largest error source in the radiative transfer simulations is the surface albedo. Surface albedo input is taken from the OMI/SCIAMACHY surface LER database. To study the inevitable errors brought about by adopting non-directional Lambertian surface albedo, we study the FRESCO (Wang et al., 2008, 2012) cloud information also stored while processing the TROPOMI data. In Fig. 4 we plot the (effective) cloud fraction retrieved by the TROPOMI FRESCO algorithm for all clear-sky latitude/longitude boxes on 1 May 2018. The TROPOMI FRESCO cloud fraction from boxes located over land is plotted in green, for boxes located over the ocean blue is used. The NPP-VIIRS geometrical cloud fraction is also plotted, in red. The NPP-VIIRS cloud fraction is always close to zero, because it was used for the cloud masking. The FRESCO cloud fraction over the ocean is also close to zero, as expected. However, the FRESCO cloud fraction over land deviates quite a lot from zero.

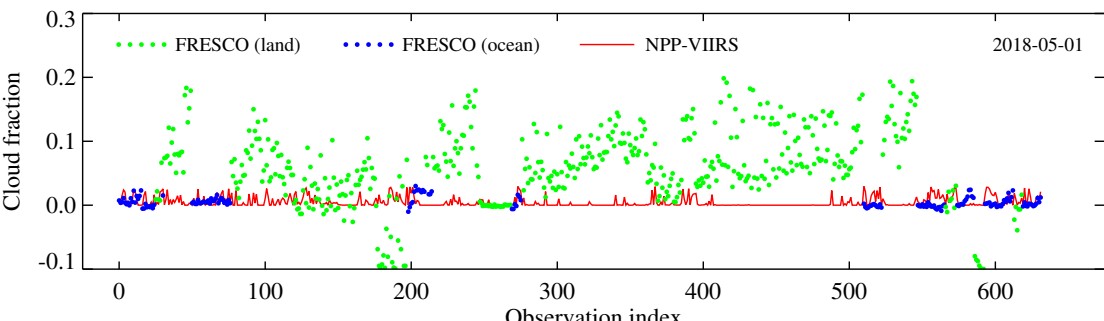

**Figure 4.** Cloud fractions calculated from TROPOMI FRESCO and NPP-VIIRS observations for all cloud-free $1° \times 1°$ latitude/longitude boxes on 1 May 2018. The FRESCO data are plotted in green over land and in blue over the ocean. The NPP-VIIRS cloud fractions, plotted in red, are all close to zero, as expected, because they were used for the cloud screening. The large deviations from zero found in the FRESCO cloud fractions over land are primarily the result of inadequate (non-directional) surface albedo information provided to the FRESCO retrieval algorithm, resulting in large errors in the FRESCO cloud fractions for certain scattering geometries.

The deviations found over land are in line with conclusions from a recent study by Lorente et al. (2018), who showed that traditional Lambertian surface albedo databases, like the ones described in Sect. 3.4, seriously underestimate the surface reflectivity for certain geometries due to the fact that surface reflectance anisotropy is not accounted for in these databases. The error made this way can be as large as a factor of two in the surface albedo over forested scenes at 772 nm. For cloud retrieval algorithms that operate in the near-infrared wavelength range, like the FRESCO algorithm which makes use of the $O_2$-A band near 760 nm, the errors in the cloud fractions can be as large as 0.2 and even higher (Lorente et al., 2018, Fig. 7b).





## 5.2 Selecting the best surface albedo method

Four methods are employed to model the surface reflectivity and/or to apply additional filtering:

1. Method 1 is the standard case with OMI/SCIAMACHY surface LER as surface albedo input.

2. Method 2 additionally performs filtering on the FRESCO cloud fraction. Observations are excluded if the absolute cloud fraction exceeds 0.06. In Sect. 5.1 it was made plausible that the deviations from zero in Fig. 4 are primarily the result of the usage of a non-directional surface LER database. Cloud fractions that deviate too much from zero are therefore assumed to be indicative for incorrect surface albedo input.

3. Method 3 uses MODIS BRDF as surface albedo input. As explained in Sect. 3.4, this is not possible for most of the 21 wavelength bands. We use MODIS band 1 (centred around 645 nm) to estimate the surface BRDF at 670 nm for the scattering geometry defined by $\theta$, $\theta_0$, and $\phi - \phi_0$ and use it in the normal baseline which expects Lambertian surface albedo input. This is not entirely correct, but at 670 nm, where the amount of Rayleigh scattering in the atmosphere is low, surface-only reflection is dominating for clear-sky scenes and the above procedure is actually quite a fair approximation.

4. Method 4 filters out cases for which the Lambertian OMI/SCIAMACHY surface LER does not agree with the directional MODIS surface BRDF. Figure 5 presents the SCIAMACHY surface LER, determined for 645 nm, versus the surface BRDF from MODIS band 1, for all cloud-free land scenes observed on 1 May 2018. The differences can be rather large and we attribute these to the non-directional nature of the SCIAMACHY surface LER database. Method 4 calculates the parameter $\delta_{645}$, which represents the difference between MODIS surface BRDF and SCIAMACHY surface LER for a given scene at 645 nm:

$$\delta_{645} = A_{\mathrm{BRDF}} - A_{\mathrm{LER}} \qquad (5)$$

As we are only interested in land scenes for which the SCIAMACHY surface LER is in reasonable agreement with the MODIS BRDF we only continue with land scenes for which we have $|\delta_{645}| \leq 0.02$. These scenes correspond to the red circles in Fig. 5.

For each of these four methods the baseline method described in Sect. 4 was put to work on a full day of TROPOMI data from 1 May 2018 and complemented with radiative transfer simulations as described in Sect. 3. The results for the 670-nm wavelength band are shown in Fig. 6. The green data points represent observations taken over land, the blue data points denote scenes containing water surfaces.

For Method 1, there seems to be a fair agreement between measurement and simulation. Pearson's correlation coefficient $r$ is 0.989, indicating high correlation. The linear fit to the data points (represented by the grey line) is quite close to the one-to-one relationship (indicated by the dotted line). However, the spread in the data points is rather large. The standard deviation of the data points w.r.t. the aforementioned linear fit amounts to $\sigma = 0.022$.

For Method 2, the scatter in the scatter plot has clearly been reduced, from $\sigma = 0.022$ to $\sigma = 0.016$. Pearson's $r$ increased slightly from 0.989 to 0.995. The effect on the coefficients of the linear fit is small but significant.



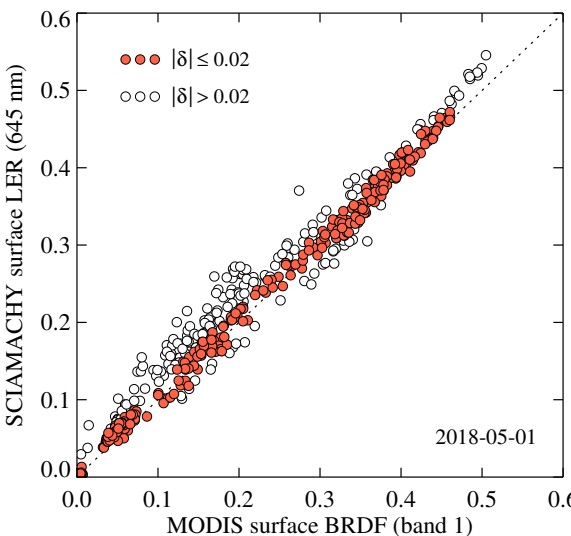

**Figure 5.** SCIAMACHY surface LER at 645 nm versus MODIS surface BRDF from MODIS band 1 (centred around 645 nm), for cloud-free conditions and land surfaces observed by TROPOMI on 1 May 2018. The red circles indicate observations for which there is good agreement ($|\delta_{645}| \leq 0.02$). These are used in Method 4.

For Method 3, the correlation between TROPOMI measurement and DAK simulation is very good. Pearson's correlation coefficient $r$ now amounts to 0.998 and the standard deviation $\sigma$ dropped down to 0.008. Note that there are no water scenes present, because MODIS BRDF is available primarily over land. If water scenes would have been removed from the Method 1 and Method 2 analyses, then their $\sigma$ values would have increased. The improvement of Method 3 in terms of $\sigma$ is therefore

substantial. However, the slope $m = 1.031$ of the linear fit is meaningless because the BRDF from MODIS band 1 is representative for 645 nm, not for 670 nm. Using MODIS BRDF directly is therefore a good illustration of the most optimal situation that can be achieved in terms of $r$ and $\sigma$, but it is not a practical way of improving the comparison.

Finally, for Method 4, the reliability of the comparison has improved compared to Methods 1 and 2. The correlation coefficient $r$ is 0.997 and, more importantly, the standard deviation $\sigma$ went down to 0.012. These numbers are close to the numbers

found for Method 3.

## 5.3 Final method

Method 4 essentially filtered out scenes with surface scattering geometries which were not well represented by the (Lambertian) SCIAMACHY surface LER database. This is now done for all the longer wavelength bands (670 to 2314 nm). For the shorter wavelength bands (328 to 494 nm) the OMI surface LER database is used, but now we use MODIS band 3 (centred around

15 469 nm) to filter out scenes for which $|\delta_{469}| > 0.025$. We use a higher threshold value, because the difference between BRDF and LER is higher due to the fact that multiple scattering has become more important.



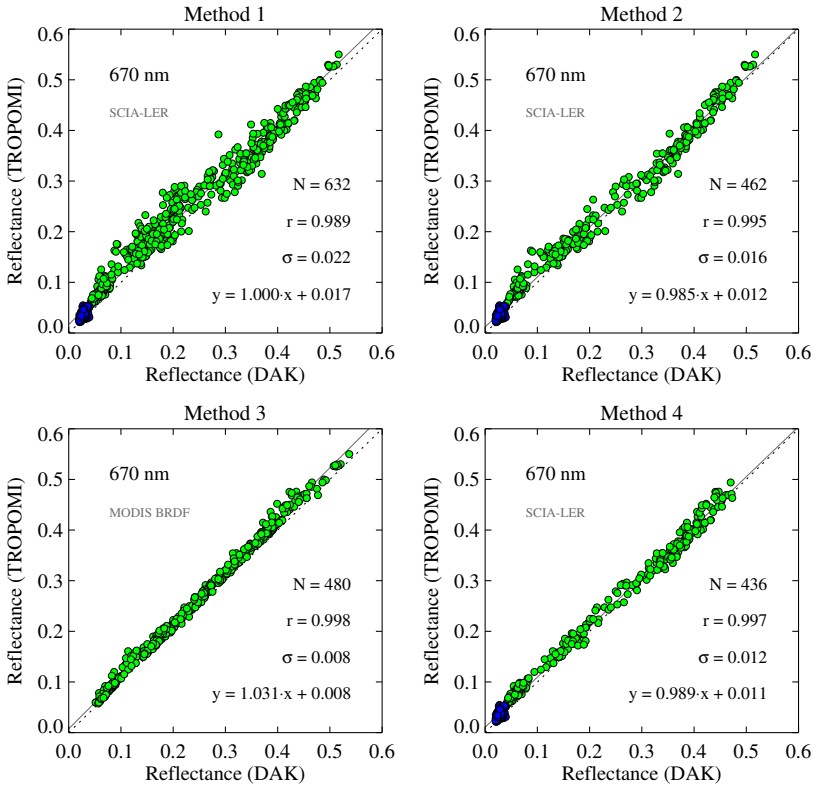

**Figure 6.** TROPOMI reflectance versus DAK simulated reflectance at 670 nm for cloud-free conditions on 1 May 2018. Four methods are employed to model the surface reflectivity and to apply filtering. Method 1 is the standard case with SCIAMACHY surface LER as surface albedo input. Method 2 is the same but with an additional filtering based on the FRESCO cloud fraction. Method 3 uses MODIS BRDF at 645 nm as albedo input. Method 4 uses SCIAMACHY surface LER but with a filtering requiring that the SCIAMACHY surface LER and MODIS BRDF at 645 nm are in good agreement. The green circles represent cloud-free scenes over land and the blue circles correspond to cloud-free water scenes. Further explanation is provided in the main text.

## 6   Results

### 6.1   First analysis

In Fig. 7 we present the results for the final method defined in Sect. 5.3 for 12 of the 21 wavelength bands. Each window contains a scatter plot with linear fit, fit coefficients, standard deviation $\sigma$, correlation coefficient $r$, and the surface LER database used. The shortest wavelength band (328 nm) shows a very low $\sigma$ value of 0.003. Here the sensitivity of the reflectance to errors in the surface reflectance is the lowest (Tilstra et al., 2005, Fig. 4). The linear fit has near zero offset ($n = 0.000 \pm 0.016$) but the slope deviates significantly from one ($m = 1.119 \pm 0.056$). As the wavelength increases, the simulations become more sensitive to errors in the surface albedo and this is reflected in the increase in the standard deviation $\sigma$, which goes up to 0.008



**Figure 7.** Reflectances measured by TROPOMI on 30 October 2019 versus simulated reflectances calculated using the DAK radiative transfer model, for 12 of the 21 wavelength bands studied in this paper. Only clear-sky observations were used in the scatter plots. As before, green circles represent clear-sky scenes over land and blue circles correspond to clear-sky water scenes. For the wavelength bands below 500 nm the surface albedo is taken from the OMI surface LER database; above 500 nm surface albedo input from the SCIAMACHY surface LER database is used. The analysis was performed using Method 4. The black lines represent linear fits $y = m \cdot x + n$ to the data. The fit results for $m$ and $n$ and the standard deviation $\sigma$ of the data points w.r.t. the linear fit are shown in each window.





at 494 nm. At this wavelength the linear fit deviates less severely from the one-to-one relationship: $m = 1.048 \pm 0.041$ and $n = -0.003 \pm 0.006$.

For the longer wavelength bands (670 to 2314 nm) the results are notably different. There is more spread in the data points, but almost near perfect correlation ($r = 0.998$ in all cases). More importantly, the one-to-one relationship is respected for all

wavelength bands except 2314 nm. The special case of 2314 nm will be discussed in Sect. 6.3. For the wavelength bands 670 to 772 nm from TROPOMI spectral bands 5/6 the slope $m$ of the linear fit is largely equal to one within the accuracy of the comparison. For example, at 670 nm, $m = 0.992 \pm 0.014$. At 758 nm, $m = 1.010 \pm 0.013$. Conclusions about the calibration of TROPOMI will be drawn in Sect. 6.4.

Studying the results we found no dependence on viewing zenith angle, or on any of the other angles, for any of the 21

wavelength bands that were studied. The existence of such a dependence could have indicated problems with the radiative transfer calculations at the shorter wavelengths, where the atmospheric contribution is large, or it could have pointed to errors introduced mainly for the longer wavelengths by the handling of surface reflection in the simulations. This suggests that the comparisons that were performed are sound.

## 6.2   Time series and trends

To study whether the results are representative we repeat the analyses for the period 2018–2020 by processing one day per fortnight. This results in 56 days to be analysed. This time we determine the fit coefficients of the linear fit through the data points in the scatter plots, as before, but also the average difference between TROPOMI and the DAK simulations. The results are shown in Fig. 8 for 5 wavelength bands. The first column shows the slope $m$ of the linear fit, the second column the intercept $n$ of the linear fit, and the third column the average difference $d_{\mathrm{m}}$ between the TROPOMI and DAK reflectances. Data from

the commissioning phase are shown in red and do not contribute in any way to the analyses.

At 328 nm a small upward trend is visible in both the slope $m$ and intercept $n$ of the linear fit and even more clearly in the parameter $d_{\mathrm{m}}$. The value of $d_{\mathrm{m}}$ reaches 0.035, which is quite a large difference considering the fact that the typical value of the reflectance at this wavelength is about 0.3 (see Fig. 1). The upward trend in $d_{\mathrm{m}}$ goes down as the wavelength increases, until 494 nm, where it is nearly completely absent. At 494 nm there is a small downward trend in the slope $m$, but the small

upward trend in the intercept $n$ is comparable to the 328-nm case. The slope $m$ is, apparently, changing with wavelength, but the intercept $n$ is not.

For the longer wavelength bands no clear trends are visible. At 670 nm the average slope $\overline{m}$ is very close to one ($\overline{m} = 0.998 \pm 0.009$) but the average intercept $\overline{n}$ suggests a slight offset ($\overline{n} = 0.010 \pm 0.003$). The offset is present in both the land and sea portions of the data, and in Fig. 6 it was also present when MODIS surface BRDF was used as surface albedo input

for the 670-nm wavelength band. As a result, the offset does not seem to be caused by surface type and/or errors in the surface albedo. At 772 nm the average slope is larger than one ($\overline{m} = 1.033 \pm 0.009$) but the intercept is now very close to zero ($\overline{n} = 0.003 \pm 0.002$). The lower value of the average intercept $\overline{n}$ as compared to the 670-nm wavelength band shows that the offset found at 670 nm cannot be related to residual clouds.



**Figure 8.** Time series of the slope $m$ (left column) and intercept $n$ (middle column) of the linear fit to the data in scatter plots such as shown in Fig. 7, for 5 of the 21 wavelength bands defined in this paper. The last column presents the mean differences $d_\mathrm{m}$ between the TROPOMI and DAK reflectances. The red data points are based on data from the commissioning phase. These are plotted but not taken into account in the linear fits. In all cases the horizontal dotted line represents the situation of perfect agreement between measurement and simulation. The blue lines/curves are fits to the data and are discussed in Sect. 6.2. The blue circle illustrates the fit result for 30 April 2018.





For the wavelength band at 2314 nm the results are very different. The time series of the intercept $n$ shows no clear trend and there is only a modest average offset ($\overline{n} = 0.009 \pm 0.004$). The slope $m$, however, shows a seasonal cycle with a period $T$ of one year. We therefore fit a function of the following form to the data points:

$$m(t) = a_0 + a_1 t + a_2 \sin(2\pi t/T + a_3) \; , \tag{6}$$

where $t$ is the time in years. The fit results indicate only a negligible trend: $a_1 = 0.0004 \pm 0.0049$ yr$^{-1}$. The fitted function is plotted in blue in Fig. 8 for the 2314 nm case. The behaviour of this wavelength band will be discussed further in Sect. 6.3.

## 6.3 SWIR channel (2314 nm)

In Fig. 9 we present scatter plots of TROPOMI reflectance versus DAK reflectance for two days: 16 January 2019 (left column) and 13 June 2019 (right column). Focusing for the moment only on the scatter plots in the top row of Fig. 9, we see that for the first day the data are reasonably close to the one-to-one line. For the second day the data are significantly deviating from the one-to-one relationship. The two days illustrate the situation near the top and near the bottom of the sinusoidal dependence shown in Fig. 8 for the 2314-nm wavelength band. The question to address is whether the sinusoidal behaviour is originating from the TROPOMI data or from the radiative transfer modelling.

In-flight monitoring results for the TROPOMI SWIR channel have appeared in a recent paper by van Kempen et al. (2019). This paper reports a radiometric stability of the SWIR channel within the 0.1% level, a result which was based on analyses of TROPOMI's DLED and WLS observations. These calibration monitoring measurements are in fact representative for the Earth radiance light path. We found the solar irradiance to also be stable within the 0.1% level (not shown). The conclusion is that the sinusoidal behaviour found in Fig. 8 cannot be not caused by TROPOMI. It must therefore be caused by the radiative transfer calculations. The most logical explanation would then be that there is an issue with the 2314-nm wavelength band of the SCIAMACHY surface LER database. This would immediately explain the period $T$ of a year that was reported in Sect. 6.2.

The 2314-nm wavelength band of the SCIAMACHY surface LER database was retrieved from SCIAMACHY band 8. This band was troubled by the build up of an ice layer on the detector, and by the many attempts to remove this ice layer by heating up the detector (Lichtenberg et al., 2006). This had an impact on the throughput of the detector. However, in deriving the SCIAMACHY surface LER database, necessary corrections were applied to the 2314-nm reflectances. These corrections cannot possibly have introduced the sinusoidal behaviour, and the ice-layer related problems themselves are of a completely different nature than the clear sinusoidal relationship that is found in Fig. 9. However, there could be other causes for errors in the 2314-nm SCIAMACHY surface LER. Proof that the SCIAMACHY surface LER database is responsible for the sinusoidal behaviour was not found.

Another potential explanation for the sinusoidal behaviour could be that the approach of filtering out geometries according to Method 4 (see Sect. 5.2) breaks down at 2314 nm. To test this hypothesis, we also performed the radiative transfer calculations using MODIS surface BRDF from band 7 (centred around 2.1 µm) as surface albedo input for the radiative transfer calculations. The results are shown in the bottom row of Fig. 9. Obviously, the surface reflectivity at 2.1 µm is somewhat different from that at 2314 nm, so a clear one-to-one relationship cannot be expected. But comparing the scatter plots of the upper and lower row

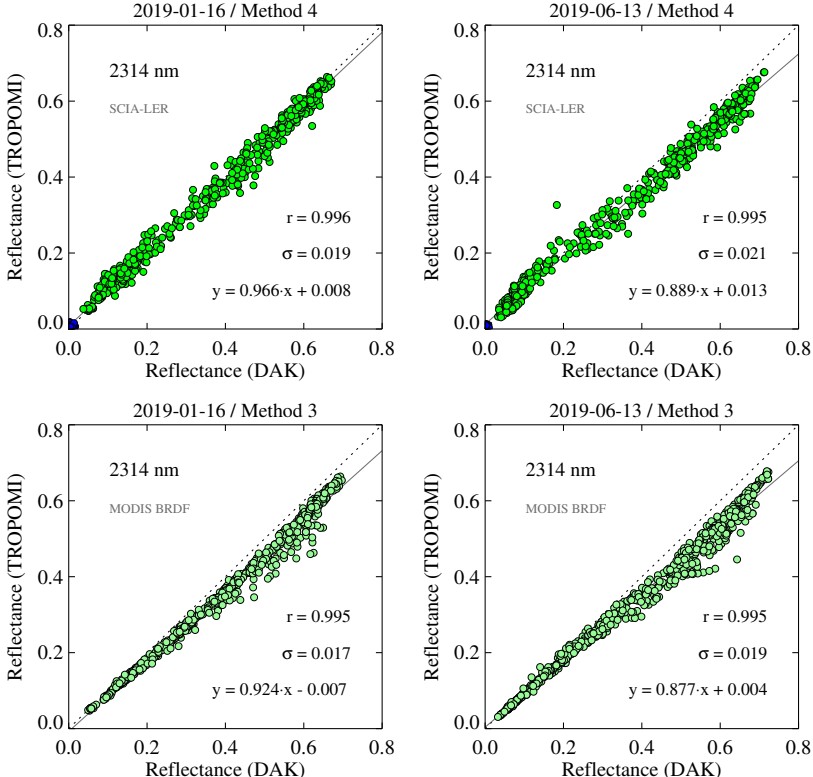

**Figure 9.** Scatter plots for the 2314-nm wavelength band for 16 January 2019 (left column) and 13 June 2019 (right column). In the top row the surface albedo used is that of the SCIAMACHY surface LER database, and filtering is applied according to Method 4. For 13 June 2019 the simulated DAK reflectance overestimates the TROPOMI reflectance much more than it does for 16 January 2019. In the bottom row the surface albedo is that of band 7 of the MODIS surface BRDF database (central wavelength 2.1 μm). It seems that using MODIS surface BRDF also leads to a larger overestimation of the simulated DAK reflectance for 13 June 2019 than for 16 January 2019.

it can be seen that in both cases the DAK simulations overestimate the reflectance compared to TROPOMI more for 13 June 2019 than for 16 January 2019. This suggests that the sinusoidal behaviour is (1) not caused by errors in the SCIAMACHY surface LER database and (2) also not caused by shortcomings in the additional filtering we apply in Method 4.

Looking further for potential discrepancies in the radiative transfer calculations is also not providing us with an explanation.
In the wavelength region surrounding 2314 nm absorption by water vapour is an issue. However, the selected wavelength is precisely located in a continuum part of the spectrum. Tests involving alterations of the slit function and wavelength assignment showed that the 2314-nm wavelength is well chosen and that its reflectance is stable. Unfortunately, all of this leaves the sinusoidal behaviour unexplained.



## 6.4 Summary and interpretation of the results

We conclude that all 21 selected wavelength bands show the correct linear response to the intensity of the detected radiation. Table 3 summarises the end results for the validation of the absolute radiometric calibration. The reported slopes, intercepts, deviations, and their errors are all representative for 30 April 2018. For TROPOMI, this day is important because it is the first

5   mission day after the end of its commissioning phase. The results from the time series presented in Fig. 8 were extrapolated via the linear fits to this particular day, which is indicated by the blue circles in Fig. 8.

The last row in Table 3 presents the percentage $D_{1.0}$, where $D_{1.0}$ for an analysed day is defined as

$$D_{1.0} = 100\% \cdot (m + n - 1) \ . \tag{7}$$

In Eq. (7) the $m$ and $n$ are the slope and intercept as defined in Sect. 6.1. The percentage $D_{1.0}$ therefore represents the estimated

10   calibration error in the TROPOMI reflectance for a case with a reflectance of 1, expressed in a percentage. Contrary to the other

**Table 3.** Final results from the reflectance validation study described in this paper. For each of the 21 wavelength bands the differences w.r.t. simulations are given. The slopes, intercepts, differences, and their uncertainties are representative for the clear-sky scenes that were studied and best describe the situation for 30 April 2018, the first day after the end of the TROPOMI commissioning phase. The percentage $D_{1.0}$ deviates from this in that it was determined for the entire time period that was studied (for which it is representative).

| Wavelength band (nm) | 328 | 335 | 340 | 354 | 367 | 380 | 388 | 402 | 416 | 425 | 440 |
|---|---|---|---|---|---|---|---|---|---|---|---|
| Spectral band | 3 | 3 | 3 | 3 | 3 | 3 | 3 | 4 | 4 | 4 | 4 |
| Surface albedo input | OMI | OMI | OMI | OMI | OMI | OMI | OMI | OMI | OMI | OMI | OMI |
| Slope | 1.086 | 1.072 | 1.057 | 1.063 | 1.059 | 1.077 | 1.061 | 1.065 | 1.072 | 1.070 | 1.078 |
| Slope uncertainty | 0.025 | 0.024 | 0.024 | 0.027 | 0.030 | 0.034 | 0.036 | 0.040 | 0.043 | 0.044 | 0.042 |
| Intercept | -0.002 | -0.001 | -0.002 | -0.004 | -0.004 | -0.005 | -0.006 | -0.008 | -0.008 | -0.008 | -0.009 |
| Intercept uncertainty | 0.007 | 0.008 | 0.008 | 0.008 | 0.008 | 0.008 | 0.008 | 0.009 | 0.008 | 0.008 | 0.007 |
| Difference | 0.021 | 0.020 | 0.015 | 0.012 | 0.010 | 0.012 | 0.007 | 0.005 | 0.005 | 0.004 | 0.004 |
| Difference uncertainty | 0.002 | 0.002 | 0.002 | 0.002 | 0.002 | 0.002 | 0.002 | 0.002 | 0.002 | 0.002 | 0.002 |
| Percentage $D_{1.0}$ | 10.0% | 8.7% | 7.2% | 7.3% | 6.9% | 8.3% | 6.5% | 6.3% | 6.6% | 6.2% | 6.5% |

| Wavelength band (nm) | 463 | 494 | | 670 | 685 | 697 | 712 | 747 | 758 | 772 | 2314 |
|---|---|---|---|---|---|---|---|---|---|---|---|
| Spectral band | 4 | 4 | | 5 | 5 | 5 | 5 | 6 | 6 | 6 | 7 |
| Surface albedo input | OMI | OMI | | SCIA | SCIA | SCIA | SCIA | SCIA | SCIA | SCIA | SCIA |
| Slope | 1.082 | 1.072 | | 0.997 | 0.999 | 0.992 | 1.002 | 1.015 | 1.019 | 1.034 | 0.917 |
| Slope uncertainty | 0.039 | 0.036 | | 0.009 | 0.009 | 0.010 | 0.009 | 0.009 | 0.008 | 0.008 | 0.023 |
| Intercept | -0.008 | -0.005 | | 0.010 | 0.010 | 0.010 | 0.007 | 0.006 | 0.003 | 0.003 | 0.011 |
| Intercept uncertainty | 0.006 | 0.005 | | 0.003 | 0.003 | 0.003 | 0.002 | 0.003 | 0.002 | 0.002 | 0.004 |
| Difference | 0.005 | 0.006 | | 0.010 | 0.010 | 0.009 | 0.008 | 0.010 | 0.007 | 0.011 | -0.011 |
| Difference uncertainty | 0.002 | 0.002 | | 0.002 | 0.002 | 0.002 | 0.002 | 0.002 | 0.002 | 0.002 | 0.005 |
| Percentage $D_{1.0}$ | 6.8% | 6.0% | | 0.7% | 1.0% | 0.6% | 1.0% | 2.2% | 2.3% | 3.8% | -5.8% |





results presented in Table 3, we did not make $D_{1.0}$ representative for 30 April 2018, but instead calculated $D_{1.0}$ for the entire time period that was studied. The $D_{1.0}$ therefore also include the effects of instrument degradation since 30 April 2018.

We can now interpret the results presented in Table 3 and draw conclusion about the radiometric calibration of the TROPOMI Earth reflectance. We start with the selected wavelength bands in TROPOMI spectral bands 5 and 6 (wavelength bands 670 to

772 nm). Here the accuracy of the comparison is estimated to be about 2–3%. We see that there is full agreement within the reported errors for all wavelength bands except at 772 nm. Here $D_{1.0}$ exceeds the accuracy of the comparison. We therefore conclude that the TROPOMI radiometric calibration is not meeting the TROPOMI calibration requirement of 2% at 772 nm. For the other wavelength bands the radiometric calibration seems to be correct within 2%.

For the selected wavelength bands in TROPOMI spectral bands 3 and 4 (wavelength bands 328 to 494 nm) we find, depend-

ing on the wavelength, differences $D_{1.0}$ between 6% and 10%. The differences are much larger than the estimated accuracy of the method (1–3%), and therefore significant. The differences thus are to be interpreted as calibration errors in the TROPOMI reflectance. It should be noted that the magnitude of the errors is in agreement with radiometric calibration errors found recently in the TROPOMI solar irradiance product (Ludewig et al., 2020, Fig. 19). We conclude that for spectral bands 3 and 4 the radiometric calibration does not meet the TROPOMI calibration requirement of 2%.

The situation for the SWIR channel is described in Sect. 6.3. Based on the discussion in this section, we have to dismiss the results for the 2314-nm wavelength bands. However, we do believe that there are indications that the reflectance of TROPOMI spectral band 7 is too low, and that further checks are needed.

Detailed end results are provided in Table 3.

## 7 Conclusions

In this paper we studied the quality of the radiometric calibration of the TROPOMI instrument by comparing the reflectances observed by TROPOMI with reflectances simulated by the radiative transfer code DAK. This was done for clear-sky scenes, to avoid the (complicated) modelling of scenes containing clouds. Comparisons between satellite observations and radiative transfer calculations have a limited accuracy, for a variety of reasons. For a large part of the wavelength range that was studied in this paper (328–2314 nm), the limiting factor in the comparison is the knowledge of the surface albedo.

There are a number of surface albedo databases available which provide the spectral surface reflectivity information needed as input for the radiative transfer calculations. However, these databases do not provide surface BRDF, but Lambertian, non-directional surface reflectance information. The established MODIS surface BRDF product provides an excellent kernel-based expansion of the surface BRDF, but only for a few wavelength bands. The MODIS wavelength bands cannot be used for the intercomparison in this paper.

In this paper we have tried to increase the quality of the radiative transfer calculations by using a mixed approach in which we use OMI and SCIAMACHY surface LER as surface albedo input and apply filtering by comparing the surface LER values with MODIS surface BRDF at a specific reference wavelength. This way, the radiative transfer simulation can make use of





the spectral surface reflectivity information that is available and at the same time take advantage of the directional information contained in the MODIS surface BRDF product. The result is a higher accuracy for the intercomparison.

From the intercomparison between TROPOMI and the radiative transfer code DAK we can conclude that all 21 selected wavelength bands show the correct linear response to the intensity of the detected radiation. Especially the reflectances in

spectral bands 5 and 6 show a good absolute agreement with the simulations. The radiometric calibration errors are on the order of 0–2%, within the accuracy estimate of the intercomparison, except for the 772-nm wavelength band, which shows larger deviations from the simulations. For this wavelength band there is a mild, but statistically significant, calibration issue.

The results for TROPOMI spectral bands 3 and 4 point to more severe problems in the radiometric calibration. For the selected wavelength bands (328 to 494 nm) we found differences with the simulations between 6% and 10%. The differences

are larger than the estimated accuracy of the method (1–3%), and therefore clearly point to calibration errors leaking into the TROPOMI reflectance. In any case, for spectral bands 3 and 4 the radiometric calibration does not meet the TROPOMI requirement of 2%.

The analysis of the 2314-nm wavelength band located in the SWIR channel showed unexpected behaviour, which we attributed to the radiative transfer calculations. This means that we have to dismiss the results for TROPOMI spectral band 7.

More research is needed to investigate the radiometric calibration of this spectral band, and of TROPOMI spectral bands 1 and 2.

*Author contributions.* LGT wrote the manuscript and performed most of the work. MdG and PW contributed to the cloud and aerosol screening. PS and PW helped with the radiative transfer modelling. All authors discussed the results and commented on the manuscript.

*Competing interests.* The authors declare that they have no conflict of interest.

*Disclaimer.* The results presented in this paper are based on version-1.0 Sentinel-5 Precursor data. All conclusions that were drawn in principle only apply to version 1.0 of these data.

*Acknowledgements.* The work presented in this paper was performed in the context of the Sentinel-5 Precursor Validation Team (S5PVT). We acknowledge the free use of TROPOMI data provided by ESA. The MODIS MCD43C1 data product was retrieved from the online Data Pool, courtesy of the NASA Land Processes Distributed Active Archive Center (LP DAAC), USGS/Earth Resources Observation and

Science (EROS) Center, Sioux Falls, South Dakota, https://lpdaac.usgs.gov/tools/data-pool/.



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
