# Peer review of "In-orbit Earth reflectance validation of TROPOMI on board the Sentinel-5 Precursor satellite"

_Atmospheric Measurement Techniques, 2020_

## Referee Comment (RC1) · Anonymous Referee #1 · 8 May 2020

I wish to thank the authors for a well-written paper. It is structured and organized, making it easy to follow. In most cases the descriptions are precise, so that the meaning is not left to interpretation by the reader. The conclusions are reasonable; the authors do not over-interpret their results.

Following are several issues I would like addressed.

Section 1 The authors state their objective is to evaluate TropOMI radiometric accuracy using the best available surface albedo data sets. This implies they wish to address its absolute accuracy and not merely the TropOMI calibration relative to that of OMI and SCIAMACHY. Given this broader objective there should be an assessment of OMI/SCIA radiometric accuracy, and a discussion of how representative the respective LER data sets are of the underlying calibrations (an accurate calibration does not

necessarily mean the LER data sets are equally accurate). Section 5 discusses one deficiency of these LER databases, but it would be better if this paper addresses data set accuracy in Section 3 rather than as only a data screening issue in Section 5.

Section 3.2 No mention is made of Raman scattering. Should the readers assume that it is not modeled? Since it is an important effect for evaluating the spectral radiometric response of the instrument (as much as 1-2% at TropOMI wavelengths shorter then 400 nm), perhaps it should be stated explicitly.

Section 5.1 The information provided in this section is not entirely clear. It is not immediately obvious that the FRESCO false cloud identification over land is caused by an overestimation of surface reflectivity in the underlying databases. Also, the underestimation of land reflectivity, while an established effect, has not been established for these particular databases as their sole or even primary error. In many locations the OMI LER dataset values are higher than other standards, presumably as a result of sub-pixel cloud contamination. I will presume that SCIAMACHY LER suffers similarly.

Section 5.2 The alternative methods presented in this section appear to be designed to eliminate scenes for which the LER databases are affected by non-Lambertian surface characteristics. Since these errors are as much a function of viewing conditions as they are surface type, isn't the TropOMI LER subject to similar errors? Is it legitimate to screen the reference data and not the measured data as well?

Section 6.2 It is not clear where the referenced 772 nm slope and intercept are coming from. Is it Figure 6 or Figure 8?

---

## Referee Comment (RC2) · Ruediger Lang (Referee) · 15 May 2020

The paper by Tilstra et al. on the In-orbit Earth reflectance validation of TROPOMI on board the Sentinel-5 Precursor satellite is an important contribution to the meanwhile significant history of knowledge in characterising this class of instruments, since the launch of GOME-1 on ERS-2. The paper is well written and organized and it shows robust and convincing results on the assessment of the TropOMI instruments radiometric in-flight calibration accuracy using independent radiative transfer (RT) forward modelling of the expected Top-Of-Atmosphere (TOA) radiances in the region between 328 and 2314 nm.

The fact that the results are convincing and robust with respect to the provided sta-

tistical analysis of the reflectance correlations (between simulated and measured results) is not a given, since previous attempts to use RT model results in evaluating the in-flight calibration accuracy and performance of this class of sensors were limited predominantly by uncertainties in model inputs.

In this respect the statistical significance of the presented results, and the fact that they may be used as a robust evaluation of the sensor performance for most of the large spectral regions investigated here, makes it a unique contribution, generally in the field of post-launch high-spectral resolution sensor calibration in the UV to the near infra-red (NIR).

In this respect, and in view of future missions in development (Sentinel-5), or planned mission like the proposed High-Priority Copernicus Candidate $CO_2$ monitoring mission (HPCC CO2M), it is unfortunate that the short-wave infra-red (SWIR) band evaluation had to be excluded from the results, probably because of systematic error contributions in the RT model surface reflectance input. In view of the significantly stronger focus of S5 and CO2M on the SWIR, improving the analysis and associated methods in this spectral region would be very welcome (and as suggested by the authors they consider this possible provided a better surface albedo input becomes available).

The authors address the two key error-sources for the overall approach in quite some detail. One is the selection of real clear-sky Rayleigh scattering scenes, with no or negligible residual cloud or aerosol scattering contribution. The other is the accuracy of the used surface reflection data, and in particular their angular and spectral variation (spectral BRDF), which is a large contributor - in addition to the accurate knowledge of the surface albedo itself - to the total budget of the forward modelling RT error and has significantly hampered previous attempts for carrying out such studies.

Cloud screening is predominantly done using co-located VIIRS cloud detection, and aerosols are detected by deriving an Aerosol Absorbing Index (AAI) from the measurements themselves. Here the authors state that they avoid a vicious circular problem by

"correcting" the reflectances. However it seems not quite clear if the way these valuse are corrected is then actually implying an overall iterative approach, i.e. going through the full procedure, coming up with an radiometric offset (or residual between measurement and RT result), then use these results to correct the radiances, and finally repeat the procedure until some convergence criterion is fulfilled. This aspect should be, from my point of view, better described or clarified in the paper.

The accuracy of the used surface reflectance is for sure the most critical aspect for the performance of the RT forward model results in the described validation approach. The author's present 4 methods to validate the Lambertian Equivalent Reflectance (LER) database derived from OMI and SCIAMACHY, the latter being probably used in order to cover the SWIR. The missing angular variation is considered as one of the largest errors in the existing OMI and SCIA LER databases, especially over vegetation, and this is why the chosen filtering on low quality LER values makes use of BRDF and albedo information as provided by Modis the Terra or Aqua satellite platforms. This however raises two questions: 1) Since the problem in the missing angular information in the LER has already been described for the existing LER databases derived from GOME-2, and therefore an angular dependent version of the GOME-2 LER database has been developed in the context of the Atmospheric Composition Satellite Application Facility (ACSAF) activities for Metop, the question arises why this database has not been used here. Especially since, in the end, the SCIA LER database in the SWIR shows deficiencies, and the analysis of this region had to be excluded anyway. 2) In the context of TOA test-data simulations of this kind of spectrometers, the MODIS surface albedo and BRDF is frequently used in conjunction with MODIS surface type characterisation and the ADAMs (A surface reflectance Database for ESA's earth observation Missions - https://nebula.esa.int/sites/default/files/neb_study/1089/C4000102979ExS.pdf) spectral database, which provides the possibility to calculate the angular dependent BRDF at any wavelength from the UV to the SWIR (using principle components of spectral vectors for various surface types). This proved to provide realistic BRDF values in the wavelength region covered here in simulation studies for that type of sensors. Why has

this option not been considered? Since this approach might turn out to be useful to solve the issue for the SWIR band radiometric performance validation for TropOMI.

I am sure the authors will have some convincing answers on the few issues I have raised here, in which case I can highly recommend the paper for publication in AMT.

Minor and editorial issues:

p.2, l. 13: The reference to GOME-2 on Metops could be associated to the relevant paper by Munro et al.

General, the exclusion of band 8 should probably be motivated more towards the beginning of the paper.

Section 3.4. The temporal aspects on using a database derived from SCIAMACHY and its application to a recent missions, should probably be mentioned and/or addressed, especially for vegetation and crop surfaces.

p. 8, l.3. "From Fig. 1 it can be inferred. . .". I find it actually quite difficult to infer it from the Figure if the differences are small. From the Figure 1 one can only for sure infer that they are in the right overall magnitude and spectral relation.

p. 10, l.17ff: How is the inhomogeneity of the target area actually determined? Is it just derived from coastline maps and surface type database or from the TropOMI radiance variance itself - which would be the best option I guess? In this respect, can the surface albedo as derived from the OMI and SCIA databases can be considered a true average over the 1 by 1 box as used here as a target?

Figure 3 shows the location and number of measurements over the defined period and number of days for clear sky scenes. However, it is not clear if this is the final statistics for all 56 days applying method 4 for cloud screening. The distribution of the locations of the latter, which actually go into the results, would here be of highest interest.

---

## Author Comment (AC1) · 12 Jun 2020

**Response to Reviewer 1:**

We would like to thank the reviewer for performing a thorough review and for the many helpful suggestions to improve the paper.

Below, we respond to each of the review comments. For the sake of clarity, the review comments are given in blue italics and our response is printed in normal font. Changes to the manuscript are printed in green.

*I wish to thank the authors for a well-written paper. It is structured and organized, making it easy to follow. In most cases the descriptions are precise, so that the meaning is not left to interpretation by the reader. The conclusions are reasonable; the authors do not over-interpret their results.*

*Following are several issues I would like addressed.*

*Section 1 The authors state their objective is to evaluate TropOMI radiometric accuracy using the best available surface albedo data sets. This implies they wish to address its absolute accuracy and not merely the TropOMI calibration relative to that of OMI and SCIAMACHY. Given this broader objective there should be an assessment of OMI/SCIA radiometric accuracy, and a discussion of how representative the respective LER data sets are of the underlying calibrations (an accurate calibration does not necessarily mean the LER data sets are equally accurate). Section 5 discusses one deficiency of these LER databases, but it would be better if this paper addresses data set accuracy in Section 3 rather than as only a data screening issue in Section 5.*

Indeed, our goal is to assess – as much as possible – the absolute accuracy of the radiometric calibration, and not just to present a comparison relative to that of OMI and SCIAMACHY. This implies the use of a radiative transfer code to minimise the dependence on external information. Nevertheless, this approach still relies on a good knowledge of the input parameters. The most important input parameter in the wavelength range that was studied is surface albedo. The surface LER databases that were used have been validated extensively in the past, with known, citeable accuracies. These are now mentioned in Section 3.4 of the manuscript:

" The OMI and SCIAMACHY surface LER databases have been compared to each other and to other surface LER databases [Kleipool et al., 2008; Tilstra et al., 2017]. For the OMI surface LER database an overall accuracy of 0.01–0.02 was reported, with slightly increasing values towards the shorter wavelengths [Kleipool et al., 2008]. The SCIAMACHY surface LER database was shown to have an accuracy of about 0.01 for the UV-VIS-NIR spectral range [Tilstra et al., 2017]. These accuracies reflect the uncertainties caused by various error sources, such as errors in the radiometric calibration and the occurrence of cloud contamination in the databases. Errors brought about by the Lambertian-equivalent nature of the databases are not part of these uncertainties. It should also be noted that the OMI and SCIAMACHY surface LER databases are mostly representative for the time periods from which they were derived (OMI: 2005–2009; SCIAMACHY 2002–2012). Systematic changes in surface reflectivity occurring after these time periods, for instance due to changes in land use, are not covered by the databases and will result in errors. "

*Section 3.2 No mention is made of Raman scattering. Should the readers assume that it is not modeled? Since it is an important effect for evaluating the spectral radiometric response of the instrument (as much as 1-2% at TropOMI wavelengths shorter then 400 nm), perhaps it should be stated explicitly.*

Raman scattering is not modelled. The DAK RTM that we use does not have the possibility to do so. But, the wavelength bands that are defined in Table 2 have a typical bandwidth of one nm in the UV wavelength range. Raman effects in the TROPOMI reflectance spectra are therefore averaged out partly by the spectral averaging that takes place in the conversion from reflectance spectra to reflectance bands.

This should indeed have been mentioned in of the paper. In the revised manuscript the following text is added to the manuscript, in Section 3.2:

" Raman scattering is not modelled by DAK. However, Raman effects in the TROPOMI reflectance spectra are averaged out partly by the spectral averaging that takes place in the conversion from reflectance spectra to reflectance bands. "

*Section 5.1 The information provided in this section is not entirely clear. It is not immediately obvious that the FRESCO false cloud identification over land is caused by an overestimation of surface reflectivity in the underlying databases. Also, the underestimation of land reflectivity, while an established effect, has not been established for these particular databases as their sole or even primary error. In many locations the OMI LER dataset values are higher than other standards, presumably as a result of sub-pixel cloud contamination. I will presume that SCIAMACHY LER suffers similarly.*

Yes, this is true. On the one hand, the surface albedo retrieved for the surface LER databases is systematically underestimated for certain geometries, because the algorithms are searching for the minimum scene LER values that are observed but not taking into account the directional dependence of the surface reflection. On the other hand, the surface albedo can be overestimated as a result of cloud contamination in the databases. The OMI and SCIAMACHY surface LER databases both suffer from such issues.

The text in Section 5.1 was rewritten to make it more clear that the FRESCO false cloud identification is caused by inaccuracies in the input surface albedo:

" The deviations in the cloud fractions over land are primarily caused by a systematic underestimation of the surface reflectivity for certain geometries in the (Lambertian) surface albedo database that is used in the FRESCO algorithm. This explanation is in line with conclusions from a recent study by Lorente et al. [2018], who showed that traditional Lambertian surface albedo databases, like the ones described in . . . "

*Section 5.2 The alternative methods presented in this section appear to be designed to eliminate scenes for which the LER databases are affected by non-Lambertian surface characteristics. Since these errors are as much a function of viewing conditions as they are surface type, isn't the TropOMI LER subject to similar errors? Is it legitimate to screen the reference data and not the measured data as well?*

It would not be legitimate, but it is not done that way. The filtering is performed on a box-to-box basis. That is, if the input surface albedo for a one-by-one degree box is rejected, then so is the associated TROPOMI observation box. The measured and reference data are therefore subjected to the same screening mask.

*Section 6.2 It is not clear where the referenced 772 nm slope and intercept are coming from. Is it Figure 6 or Figure 8?*

The analysis that is used to determine the average slopes and intercepts is visualised in Figure 8, but not for the 772-nm wavelength band.

This is indeed confusing, so we have changed the manuscript. We now also mention the result for 758 nm in the text, because this wavelength band is shown in Figure 8. We still mention the 772-nm results. Mentioning the 772-nm wavelength band is important, because it shows the largest deviation. In the manuscript we have added the comment that the 772-nm results are not shown in Figure 8. The changes to the manuscript are:

" ... The results were obtained for all 21 wavelength bands. In Fig. 8 the results are presented for 5 of the wavelength bands. ... "

" ... At 758 nm the average slope is larger than one ($\overline{m} = 1.015 \pm 0.008$) and the intercept is small ($\overline{n} = 0.004 \pm 0.003$). At 772 nm (not shown in Fig. 8) the average slope has increased further ($\overline{m} = 1.033 \pm 0.008$) and the intercept is now very close to zero ($\overline{n} = 0.003 \pm 0.002$) ... "

**Changes to the manuscript:**

During the review phase of this manuscript the TROPOMI instrument kept generating new data and we took the opportunity to extend the time range that was studied originally (May 2018–February 2020) by three months. The studied time period is now May 2018–May 2020, covering two years. Some of the numbers reported in the paper have changed slightly, but not significantly. Figure 8 and Table 3 have also been updated. The changes w.r.t. the previous version are in all cases insignificant, well within the reported accuracies. The extension of the time range that was studied has led to an increased accuracy of the results. Figure 3 was also updated.

**References:**

Kleipool, Q. L., Dobber, M. R., de Haan, J. F., and Levelt, P. F.: Earth surface reflectance climatology from 3 years of OMI data, J. Geophys. Res., 113, D18308, doi:10.1029/2008JD010290, 2008.

Lorente, A., Boersma, K. F., Stammes, P., Tilstra, L. G., Richter, A., Yu, H., Kharbouche, S., and Muller, J.-P.: The importance of surface reflectance anisotropy for cloud and NO2 retrievals from GOME-2 and OMI, Atmos. Meas. Tech., 11, 4509–4529, doi:10.5194/amt-11-4509-2018, 2018.

Tilstra, L. G., Tuinder, O. N. E., Wang, P., and Stammes, P.: Surface reflectivity climatologies from UV to NIR determined from Earth observations by GOME-2 and SCIAMACHY, J. Geophys. Res.-Atmos., 122, 4084–4111, doi:10.1002/2016JD025940, 2017.

---

## Author Comment (AC2) · 12 Jun 2020

**Response to Reviewer 2:**

We would like to thank Dr. Lang for performing a thorough review and for the many helpful suggestions to improve the paper.

Below, we respond to each of the review comments. For the sake of clarity, the review comments are given in blue italics and our response is printed in normal font. Changes to the manuscript are printed in green.

*The paper by Tilstra et al. on the In-orbit Earth reflectance validation of TROPOMI on board the Sentinel-5 Precursor satellite is an important contribution to the meanwhile significant history of knowledge in characterising this class of instruments, since the launch of GOME-1 on ERS-2. The paper is well written and organized and it shows robust and convincing results on the assessment of the TropOMI instruments radiometric in-flight calibration accuracy using independent radiative transfer (RT) forward modelling of the expected Top-Of-Atmosphere (TOA) radiances in the region between 328 and 2314 nm.*

*The fact that the results are convincing and robust with respect to the provided statistical analysis of the reflectance correlations (between simulated and measured results) is not a given, since previous attempts to use RT model results in evaluating the in-flight calibration accuracy and performance of this class of sensors were limited predominantly by uncertainties in model inputs.*

*In this respect the statistical significance of the presented results, and the fact that they may be used as a robust evaluation of the sensor performance for most of the large spectral regions investigated here, makes it a unique contribution, generally in the field of post-launch high-spectral resolution sensor calibration in the UV to the near infra-red (NIR).*

*In this respect, and in view of future missions in development (Sentinel-5), or planned mission like the proposed High-Priority Copernicus Candidate CO2 monitoring mission (HPCC CO2M), it is unfortunate that the short-wave infra-red (SWIR) band evaluation had to be excluded from the results, probably because of systematic error contributions in the RT model surface reflectance input. In view of the significantly stronger focus of S5 and CO2M on the SWIR, improving the analysis and associated methods in this spectral region would be very welcome (and as suggested by the authors they consider this possible provided a better surface albedo input becomes available).*

It is indeed very unfortunate that no strong conclusions could be drawn from the SWIR results. It is still not understood what exactly causes the unexpected and suspicious behaviour for the 2314-nm wavelength band. Because we cannot explain the behaviour for this wavelength band we are forced to dismiss the end result for this wavelength band. We are at the same time, however, not excluding the possibility that the results that were found for the 2314-nm wavelength band are "real". Further studies will have to investigate this.

*The authors address the two key error-sources for the overall approach in quite some detail. One is the selection of real clear-sky Rayleigh scattering scenes, with no or negligible residual cloud or aerosol scattering contribution. The other is the accuracy of the used surface reflection data, and in particular their angular and spectral variation (spectral BRDF), which is a large contributor - in addition to the accurate knowledge of the surface albedo itself - to the total budget of the forward modelling RT error and has significantly hampered previous attempts for carrying out such studies.*

*Cloud screening is predominantly done using co-located VIIRS cloud detection, and aerosols are detected by deriving an Aerosol Absorbing Index (AAI) from the measurements themselves. Here the authors state that they avoid a vicious circular problem by "correcting" the reflectances. However it seems not quite clear if the way these valuse are corrected is then actually implying an overall iterative approach, i.e. going through the full procedure, coming up with an radiometric offset (or residual between measurement and RT result), then use these results to correct the radiances, and finally repeat the procedure until some convergence criterion is fulfilled. This aspect should be, from my point of view, better described or clarified in the paper.*

The corrections on the reflectances that are used to calculate the AAI are not completely based on the validation study, but partly derived independently.

To explain, we independently analysed the reflectances at 340/380 nm which are stored in the TROPOMI AAI product. From this, a (time dependent) correction for degradation was derived for the reflectances following the method explained in Tilstra et al. [2012]. This correction for degradation was applied to the TROPOMI 340/380-nm reflectances that go into our calculation of the AAI. This part of the correction does not depend on the outcome of the study, because it is derived independently.

A day-zero correction was also applied to the 340/380-nm reflectances, by focusing on one day only ("day zero", the first day studied after the end of the commissioning phase, 1 May 2018, indicated by the blue dots in Figure 8). For this particular day, the reflectance correction factors were found from the reflectance validation study and these were then used as day-zero corrections for the reflectances that go into the AAI calculation. This part of the correction *does* depend on the validation study itself.

However, the correction factors resulting from the validation study are not changing significantly if a second round is performed to calculate the day-zero correction. Note that the AAI threshold value was set to a relatively high value of 2.0 index points. The impact of the filtering on AAI is therefore rather low to begin with.

In conclusion, there is no real need for an interative scheme.

Note that the primary reason for calculation the AAI ourselves is the fact that it is not allowed to average AAI values, but it is allowed to average the reflectances from which the AAI is derived, as explained in Section 4.3.

We have added the following sentences to Section 4.3 of the paper, to better explain the approach that was followed to correct the reflectances that go in to the calculation of the AAI:

" The corrections consist of a time-dependent multiplicative correction to account for the linear drift observed in the 340/380-nm reflectances that are stored in the S5P AAI product, and of a day-zero multiplicative reflectance correction. The linear drift was determined using the method presented in Tilstra et al. [2012]. The day-zero correction factors were found from applying the reflectance validation study on 1 May 2018 ("day zero") and determining the correction factors found for 340 and 380 nm. In principle this step could be repeated by applying the correction factors, calculating newer correction factors from these, until convergence is reached. An iterative scheme for this was not pursued, however, because a secondary iteration turned out to already provide almost the same correction factors as the first one. "

*The accuracy of the used surface reflectance is for sure the most critical aspect for the performance of the RT forward model results in the described validation approach. The author's present 4 methods to validate*

*the Lambertian Equivalent Reflectance (LER) database derived from OMI and SCIAMACHY, the latter being probably used in order to cover the SWIR. The missing angular variation is considered as one of the largest errors in the existing OMI and SCIA LER databases, especially over vegetation, and this is why the chosen filtering on low quality LER values makes use of BRDF and albedo information as provided by Modis the Terra or Aqua satellite platforms. This however raises two questions:*

*1) Since the problem in the missing angular information in the LER has already been described for the existing LER databases derived from GOME-2, and therefore an angular dependent version of the GOME-2 LER database has been developed in the context of the Atmospheric Composition Satellite Application Facility (ACSAF) activities for Metop, the question arises why this database has not been used here. Especially since, in the end, the SCIA LER database in the SWIR shows deficiencies, and the analysis of this region had to be excluded anyway.*

The *directional* GOME-2 surface albedo database can unfortunately not be used to provide the missing angular information, because the directional information stored in this database is representative for polar orbits with a 09:30 LT equator crossing time. The GOME-2 instruments are all kept in 09:30 LT orbits, while TROPOMI is in a 13:45 LT orbit. It is only possible to use the GOME-2 directional information for data observed from polar orbits with equator crossing times close to 09:30 LT. This includes data from GOME-1, SCIAMACHY, GOME-2 itself, the future Sentinel-5/UVNS, but not data from OMI or TROPOMI.

It is possible to use the *non-directional* GOME-2 surface LER information which is also stored in this product. In fact, this is what we did during the initial phase of the study. Unfortunately, the GOME-2 surface LER has some issues for wavelengths below 400 nm. These issues have been reported in the past and one could easily take this into account via corrections. However, this would not improve the readability of the paper and it would also not improve the results of this study because the SCIAMACHY surface LER database does not have a lower quality than the GOME-2 surface LER database. SCIAMACHY does have the advantage that its equator crossing time (10:00 LT) is a bit closer to the 13:45 LT of TROPOMI.

SCIAMACHY also provides surface albedo for the SWIR channel. Although we indeed had to exclude the results for the 2314-nm wavelength band from the manuscript, we were at least able to report that linearity problems and offset issues do not seem to be present for the SWIR channel. We would not have been able to draw these conclusions using the GOME-2 surface LER database.

In the revised manuscript we now mention in Section 3.4 that the GOME-2 surface LER database could also have been used, but that we decided to use the SCIAMACHY surface LER database primarily because of the additional information in the SWIR wavelength range:

" As an alternative to the SCIAMACHY surface LER database we could also use the very similar GOME-2 surface LER database [Tilstra et al., 2017]. However, the GOME-2 surface LER database does not provide information for the SWIR wavelength range, which is the primary reason for adopting the SCIAMACHY surface LER database instead. Alternatively, we can make use of MODIS surface BRDF . . . "

*2) In the context of TOA test-data simulations of this kind of spectrometers, the MODIS surface albedo and BRDF is frequently used in conjunction with MODIS surface type characterisation and the ADAMs (A surface reflectance Database for ESA's earth observation Missions - https://nebula.esa.int/sites/default/files/*

*neb_study/1089/C4000102979ExS.pdf ) spectral database, which provides the possibility to calculate the angular dependent BRDF at any wavelength from the UV to the SWIR (using principle components of spectral vectors for various surface types). This proved to provide realistic BRDF values in the wavelength region covered here in simulation studies for that type of sensors. Why has this option not been considered? Since this approach might turn out to be useful to solve the issue for the SWIR band radiometric performance validation for TropOMI. I am sure the authors will have some convincing answers on the few issues I have raised here, in which case I can highly recommend the paper for publication in AMT.*

The possibility of using the ADAM database was considered and discussed, but in the end we did not go for this option. The principal component method used in the ADAM database & toolkit seems to be a responsible way of handling the interpolation between the MODIS wavelength bands. Certainly, the wavelength interpolation provided this way is addressing the problem that we mentioned in the manuscript (in Section 3.4) of the – for our specific purpose – inconvenient location of the MODIS wavelength bands.

However, we do have our doubts that the interpolation can be done with a high enough accuracy, because the seven MODIS bands are quite far apart. For example, the shortest MODIS wavelength band is centred at 469 nm, which would involve extrapolation (not interpolation) to get towards 328 nm. A lower quality is to be expected here, as reported in, for instance, Bacour et al. [2020]. For the longest MODIS wavelength band centred at 2130 nm we would need extrapolation to get to 2314 nm. For the wavelength region at about 700 nm, interpolation has to been done between 645 and 858 nm with no support from space-based observations. Especially here, around the red edge, interpolation of any kind seems to be challenging. The OMI and SCIAMACHY surface LER databases provide surface LER for all wavelength bands studied in our validation. The (monochromatic) wavelength bands of these databases were defined in exactly the same way as the wavelength bands defined in the manuscript.

In the recent paper by Bacour et al. [2020] the ADAM database and its toolkit are introduced extensively. The paper mentions that the ADAM database *overestimates* the BRDF near the "hot spot" by a factor of about 1.5 for barren surfaces. For vegetated and other surfaces, the ADAM database *underestimates* the BRDF near the hot spot by a factor of 1.5–2.0. This is quite a large underestimation/overestimation, and it is not clear to us whether these errors are caused by the ADAM interpolation scheme or by the underlying MODIS BRDF database. It would be an alarming fact if these errors would be present in the MODIS surface BRDF information. In any case, the reported underestimation/overestimation by a factor of 1.5–2.0 is a thing to worry about, because with the close to noon (13:45 LT) equator crossing time time of TROPOMI and the relatively wide orbit swath the backscatter condition is found in a rather large part of the TROPOMI orbit.

The water BRDF in the ADAM database is not derived from MODIS, but modelled. Although modelling surface reflection for water surfaces is perfectly possible, it could lead to a difference in quality between land and water pixels. Such differences are not found in our analyses, because we treat water and land pixels in the same way. If there are deficiencies in our approach because of issues in the OMI or SCIAMACHY surface LER used, then at least the error is made in a consistent way for both land and water surfaces.

Another thing to consider is that using ADAM surface albedo input would link the validation results to the radiometric calibration of MODIS. Right now, they are linked to OMI and SCIAMACHY. One could argue that the validation results are also linked to MODIS, because of the filtering which is partly based on MODIS.

But for the most part, the link is with OMI and SCIAMACHY. This seemed to us to be the proper thing to do, especially considering the heritage of the TROPOMI instrument and of its retrieval algorithms.

In the end, for the reasons mentioned above, we decided to rely on the OMI and SCIAMACHY databases.

In the revised manuscript we now mention the ADAM database, that this approach would address the problem of the position of the MODIS wavelength bands, and explain why we decided not to use it. The text now reads:

" Another possible source of surface albedo is the ADAM database [Bacour et al., 2020]. This database is based on MODIS BRDF, but provides interpolation to any wavelength in the wavelength range 240–4000 nm. This would address the issue related to the position of the seven MODIS wavelength bands. However, interpolation does lead to a lower quality and reliability, as explained in Bacour et al. [2020]. Next to this, the ADAM database shows both underestimation and overestimation by a factors of 1.5–2 in backscatter directions [Bacour et al., 2020, p. 18]. For water surfaces, the surface reflection is modelled, and bears no relationship with the MODIS BRDF data. This means that consistency in calibration between land and ocean is broken. For the above reasons we decided to use the OMI/SCIAMACHY approach. "

*Minor and editorial issues:*

*p.2, l. 13: The reference to GOME-2 on Metops could be associated to the relevant paper by Munro et al.*

Yes, it should indeed. We have added the reference to Munro et al. [2016] to the revision of the paper.

We have also added the reference to TROPOMI, which was also missing.

*General, the exclusion of band 8 should probably be motivated more towards the beginning of the paper.*

We agree and now do this by referring to Section 4.1 already in the Introduction of the manuscript:

" The analyses cannot be performed for TROPOMI bands 1, 2, and 8 for reasons explained in Sect. 4.1. "

*Section 3.4. The temporal aspects on using a database derived from SCIAMACHY and its application to a recent missions, should probably be mentioned and/or addressed, especially for vegetation and crop surfaces.*

We agree and have added the following lines to Section 3.4:

" It should also be noted that the OMI and SCIAMACHY surface LER databases are mostly representative for the time periods from which they were derived (OMI: 2005–2009; SCIAMACHY 2002–2012). Systematic changes in surface reflectivity occurring after these time periods, for instance due to changes in land use, are not covered by the databases and will result in errors. "

*p. 8, l.3. "From Fig. 1 it can be inferred. . .". I find it actually quite difficult to infer it from the Figure if the differences are small. From the Figure 1 one can only for sure infer that they are in the right overall magnitude and spectral relation.*

We agree and have changed the sentence:

" However, the impact is relatively small and of little concern. "

*p. 10, l.17ff: How is the inhomogeneity of the target area actually determined? Is it just derived from coastline maps and surface type database or from the TropOMI radiance variance itself - which would be the best option I guess? In this respect, can the surface albedo as derived from the OMI and SCIA databases can be considered a true average over the 1 by 1 box as used here as a target?*

The term "inhomogeneity" is meant as inhomogeneity due to the occurrence of both land and water inside the box. It is based on the surface type information that is available in the TROPOMI L2 products.

Inhomogeneity in terms of the standard deviation of the TROPOMI reflectances was calculated, but used to track down outliers which were not detected by the TROPOMI flagging. This is sometimes an issue for the SWIR channel. It was not used for filtering out scenes/boxes that were considered too inhomogeneous.

The idea behind the one-by-one degree boxes is to have the OMI and SCIAMACHY surface albedo information line up as best as possible with the TROPOMI observations.

In the revised manuscript, we now clarify how the inhomogeneity of a box is determined:

" . . . sun glint or solar eclipse events. Inhomogeneity and snow/ice presence are determined based on the surface type and surface condition indicators inside the TROPOMI L2 products. A box is only considered inhomogeneous if it contains both land and water. The number of boxes . . . "

*Figure 3 shows the location and number of measurements over the defined period and number of days for clear sky scenes. However, it is not clear if this is the final statistics for all 56 days applying method 4 for cloud screening. The distribution of the locations of the latter, which actually go into the results, would here be of highest interest.*

It is the final statistics for all 56 days, so Figure 3 illustrates the distribution and frequency of the clear-sky one-by-one degree boxes that were collected over all 56 days. These are all boxes which were not filtered away for reasons explained in the text, such as presence of snow or absolute latitudes above 60 degrees. Filtering such as performed in Method 4 was not applied for this image, because the other three methods are also discussed in the paper, and because Method 4 has not been defined at this point in the manuscript.

To clarify the meaning of Figure 3 we have changed the caption of Figure 3 in the following way:

" Figure 3. Location and frequency of the $1° \times 1°$ cloud-free latitude/longitude boxes that were selected by the algorithm every two weeks for the period from May 2018 until February 2020. A total of 56 days contributed to the image. The selection and filtering approach is described in Sect. 4.4. "

**Changes to the manuscript:**

During the review phase of this manuscript the TROPOMI instrument kept generating new data and we took the opportunity to extend the time range that was studied originally (May 2018–February 2020) by three months. The studied time period is now May 2018–May 2020, covering two years. Some of the numbers reported in the paper have changed slightly, but not significantly. Figure 8 and Table 3 have also been updated. The changes w.r.t. the previous version are in all cases insignificant, well within the reported accuracies. The extension of the time range that was studied has led to an increased accuracy of the results. Figure 3 was also updated.

**References:**

Bacour, C., Bréon, F.-M., Gonzalez, L., Price, I., Muller, J. P., Prunet, P., Straume, A. G.: Simulating multi-directional narrowband reflectance of the Earth's surface using ADAM (a surface reflectance database for ESA's Earth observation missions), Remote Sens., 12, 1679, doi:10.3390/rs12101679, 2020.

Munro, R., Lang, R., Klaes, D., Poli, G., Retscher, C., Lindstrot, R., Huckle, R., Lacan, A., Grzegorski, M., Holdak, A., Kokhanovsky, A., Livschitz, J., and Eisinger, M.: The GOME-2 instrument on the Metop series of satellites: instrument design, calibration, and level 1 data processing – an overview, Atmos. Meas. Tech., 9, 1279–1301, doi:10.5194/amt-9-1279-2016, 2016.

Tilstra, L. G., de Graaf, M., Aben, I., and Stammes, P.: In-flight degradation correction of SCIAMACHY UV reflectances and Absorbing Aerosol Index, J. Geophys. Res., 117, D06209, doi:10.1029/2011JD016957, 2012.

Tilstra, L. G., Tuinder, O. N. E., Wang, P., and Stammes, P.: Surface reflectivity climatologies from UV to NIR determined from Earth observations by GOME-2 and SCIAMACHY, J. Geophys. Res.-Atmos., 122, 4084–4111, doi:10.1002/2016JD025940, 2017.

---

## Author Response (AR2)

**Response to Reviewer 1:**

We would like to thank the reviewer for helping to improve the manuscript on the topic of Raman scattering.

Below, we respond to the review comments. As before, the review comments are given in blue italics and our response is printed in normal font. Changes to the manuscript are printed in green.

*I thank the authors for their thoughtful response to my prior comments and I accept all the revisions to the original manuscript that are now reflected in Version 3, with one exception. I do not believe the authors have adequately addressed the issue of Raman scattering in their analysis.*

*The authors do not provide an explicit error budget for their analysis, but their text indicates the main error source is 0.01 owing to the surface reflectivity database. The claim, effectively, is the analysis described in this paper contributes little to the uncertainty of their primary external input. The revised discussion on Raman scattering implies that its contribution to the overall error is much less and can be ignored. In fact, the Raman scattering contribution for a 1 nm bandwidth is approximately 0.01 at low and mid latitudes (see Fig. 1 of www.atmos-meas-tech.net/7/2897/2014/doi:10.5194/amt-7-2897-2014, for instance). At higher solar zenith angles the contribution rapidly increases beyond 0.01. Unless the central wavelengths in Table 2 of the manuscript were chosen to minimize Raman residuals (the authors do not say), the additional error could be as much as 0.01, positive or negative. Adjacent wavelengths could easily have an error of 0.01 in the opposite direction yielding a relative error of 0.02.*

*The authors should acknowledge the magnitude and characteristics of this additional error source. Since the Raman scattering error can be estimated reasonably well, they might also consider including the error in Table 2 for each band center wavelength.*

Thank you for pointing this out and for pointing us to the proper literature [Vasilkov et al., 2014, Fig. 1] that illustrates the magnitude of the error brought about by the neglect of Raman scattering in the radiative transfer calculations. Indeed, the paper by Vasilkov et al. [2014] presents the impact of Raman scattering for a slit function with a FWHM of 1 nm, which is representative for the wavelength bands we defined.

According to Figure 1 of the paper by Vasilkov et al. [2014], the error caused by neglect of Raman scattering in the radiative transfer calculations can be as large as 1% and even larger, depending on the conditions. Note that this 1% error corresponds to an error of about 0.003 because the Earth reflectance in the UV is about 0.3 for cloud-free scenes under normal conditions (see Figure 1 in the manuscript).

As for the wavelength bands in Table 2 of the manuscript, in the definition of the wavelength bands the impact of Raman scattering was not the leading criterion while determining the position and width of the wavelength bands, but strong Fraunhofer lines were avoided.

Applied changes to the manuscript:

We now mention the correct magnitude of the impact of Raman scattering, which can exceed the 1% level in the UV wavelength range. We also mention the fact that the errors can be larger for larger solar zenith angles, and that the effect can be positive or negative. Finally, in various places of the manuscript we now mention that this error source needs 
[revised manuscript text omitted]